# PAINTING WITH WORDS: ELEVATING DETAILED IMAGE CAPTIONING WITH BENCHMARK AND ALIGNMENT LEARNING

**Qinghao Ye**[*], **Xianhan Zeng**[*], **Fu Li, Chunyuan Li, Haoqi Fan**
ByteDance Research

## ABSTRACT

Image captioning has long been a pivotal task in visual understanding, with recent advancements in vision-language models (VLMs) significantly enhancing the ability to generate detailed image captions. However, the evaluation of detailed image captioning remains underexplored due to outdated evaluation metrics and coarse annotations. In this paper, we introduce DECAPBENCH along with a novel metric, DCSCORE, specifically designed for detailed captioning tasks. DCSCORE evaluates hallucinations and fine-grained comprehensiveness by deconstructing responses into the smallest self-sufficient units, termed primitive information units, and assessing them individually. Our evaluation shows that DCSCORE aligns more closely with human judgment than other rule-based or model-based metrics. Concurrently, DECAPBENCH exhibits a high correlation with VLM arena results on descriptive tasks, surpassing existing benchmarks for vision-language models. Additionally, we present an automatic fine-grained feedback collection method, FEEDQUILL, for preference optimization based on our advanced metric, showing robust generalization capabilities across auto-generated preference data. Extensive experiments on multiple VLMs demonstrate that our method not only significantly reduces hallucinations but also enhances performance across various benchmarks, achieving superior detail captioning performance while surpassing GPT-4o. We release the evaluation code and the model on Github[1].

## 1 INTRODUCTION

Vision-Language Models (VLMs) (Zhu et al., 2023; Liu et al., 2024b; Ye et al., 2023; Bai et al., 2023) have risen to prominence by integrating the strengths of pre-trained large language models (LLMs) and vision models, leveraging large-scale multi-modal corpora (Liu et al., 2024b; Dai et al., 2023; Li et al., 2024a). These models have demonstrated remarkable capabilities across a diverse array of tasks. To assess their visual understanding capability, numerous benchmarks have been developed, focusing on question-answering tasks, such as MMVet (Yu et al., 2023), MMStar (Chen et al., 2024a), and MMMU (Yue et al., 2024). However, these benchmarks often rely on manually defined queries and questions, which may only cover a limited domain and lead to biased evaluations (Chen et al., 2024a). Additionally, Chen et al. (2024a) highlights that poorly constructed questions could make the models rely more on textual knowledge from their training data, thus neglecting actual visual input.

In this context, the image captioning has been a fundamental task to evaluate the visual perception capabilities of VLMs. Yet, traditional image captioning benchmarks suffer from two significant limitations: (1) The evaluation metrics (Vedantam et al., 2015; Papineni et al., 2002; Lin, 2004; Hessel et al., 2021) are unreliable and show low correlation with human judgment and model capability, and (2) The captions are typically short and lack informative visual details, missing fine-grained descriptions. In contrast, modern VLMs are capable of generating hyper-detailed image captions rich in fine-grained visual information (OpenAI, 2024a; Liu et al., 2024b). These models can even extend and infer non-descriptive elements, which are often not covered by the conventional short ground-truth captions, leading to unsatisfying detail caption evaluation results. Additionally, many of the existing image captioning datasets (Lin et al., 2014; Sidorov et al., 2020) focus on short captions

---

[1]https://github.com/MAGAer13/DeCapBench

and have become outdated, necessitating a more rigorous evaluation framework for modern VLMs. To address these limitations, it is crucial to develop new benchmarks and evaluation metrics that align closely with human judgment and accurately reflect the advanced capabilities of modern VLMs.

In this paper, we aim to assess the capabilities of modern VLMs in producing *detailed image captions*. We introduce a novel metric, DCSCORE, and a comprehensive evaluation benchmark, DECAPBENCH, designed to address the challenges of hallucination and fine-grained comprehensiveness in image captioning. Our approach involves breaking down captions into the smallest self-sufficient units, termed *primitive information units*. This decomposition reduces ambiguity and enhances the transparency and interpretability of the evaluation process. By individually assessing these units, we can accurately measure both descriptive and non-descriptive parts of captions with fine granularity. Additionally, decomposing captions allows us to evaluate their coverage with high-quality, hyper-detailed reference captions. Our experiments reveal that DCSCORE achieves the highest consistency with human expert evaluations, outperforming all existing rule-based and model-based metrics. Furthermore, we present DECAPBENCH as a detailed captioning dataset that excels in measuring hallucination and fine-grained comprehensiveness. It demonstrates superior correlation with the VLM description tasks compared to other benchmarks such as MMVet and MMStar.

In addition, we embrace the concept of breaking down responses into primitive information units and introduce FEEDQUILL, a fine-grained feedback collection strategy for preference optimization. Specifically, we generate several candidate responses and decompose them into verifiable statements. Using open-source VLMs (Liu et al., 2024a; Chen et al., 2024b), we then validate the correctness of these statements and calculate a preference score to measure precision. To avoid bias towards overly concise responses, we also factor in the number of primitive information units as feedback signals. Leveraging proximal policy optimization (PPO) (Schulman et al., 2017), we optimize preferences using a reward model trained on the collected preference data. Extensive experiments demonstrate that FEEDQUILL consistently enhances performance across various VLM models on both comprehensive and task-specific benchmarks, significantly reducing hallucinations by 40.5% relative points in mmHal-V. Furthermore, our model not only outperforms GPT-4o in detailed image captioning but also exceeds GPT-4V in visual chatting, underscoring its potential and effectiveness.

The contribution of this work can be summarized as: (1) We present DCSCORE, a novel metric for image detail caption evaluation with both hallucination and comprehensiveness, and it achieves the highest consistency with human experts among existing caption metrics. (2) We introduce a new detailed caption benchmark DECAPBENCH for evaluating the captioning capability of modern VLMs, which has the highest correlation with human judgement on description task compared to other public benchmarks. (3) We propose a simple but effective fine-grained feedback collection method FEEDQUILL by decomposing responses into primitive information units and verify them individually, which is scalable for automatically collecting preference data. (4) Extensive experimental results demonstrate the efficacy of FEEDQUILL, showing reduced hallucinations, superior performance in visual chat compared to GPT-4v, and better detailed image captioning capabilities than GPT-4o.

## 2 RELATED WORK

**Image Captioning Evaluation Metrics**   Image captioning tasks are fundamental to visual-language understanding, as they assess a model's ability to comprehend and describe images accurately. Modern vision-language models (Ye et al., 2024; Chen et al., 2024b; Liu et al., 2024a; Bai et al., 2023) equipped with massive data pre-training, are capable of generating diverse and detailed image captions. Despite these advancements, evaluating captions accurately and comprehensively remains challenging. Traditional metrics, such as BLEU (Papineni et al., 2002), METEOR (Banerjee & Lavie, 2005), and CIDEr (Vedantam et al., 2015), leverage N-gram and lexical similarity with human-annotated captions but suffer from instability due to variability in phrasing. To address this issue, model-based metrics like SPICE (Anderson et al., 2016) and CAPTURE (Dong et al., 2024) parse captions using scene graphs to match ground-truth captions. Additionally, CLIPScore (Hessel et al., 2021) and PACScore (Sarto et al., 2023) utilize pre-trained vision-language models like CLIP (Radford et al., 2021) to measure the similarity between images and captions, as well as between generated and reference captions. Recently, researchers have leveraged the powerful zero-shot capabilities of large language models (LLMs) to prompt LLMs for assessing the alignment between model-generated and human-annotated captions (Chan et al., 2023; Lee et al., 2024; Liu et al., 2024b). Despite their potential, LLM-based evaluation methods face challenges in maintaining

objectivity and comprehensiveness, particularly in extending evaluation to aspects such as knowledge and atmosphere. To alleviate these problems, we propose DCSCORE, a novel image caption metric that evaluates image captions by incorporating both hallucination and comprehensiveness thoroughly.

**Learning from Feedback for VLMs**   Learning from feedback (Yu et al., 2024a; Sun et al., 2023; Zhou et al., 2024a;b) is a core technique in the post-training stage of vision language models (VLMs). This approach enhances model performance on various tasks, such as question answering (Yue et al., 2024; Liu et al., 2023; Chen et al., 2024a) and reducing hallucinations (Li et al., 2023b), through alignment learning techniques like PPO (Schulman et al., 2017), DPO (Rafailov et al., 2024), and RLOO (Ahmadian et al., 2024). The quality of feedback is crucial for aligning models with human preferences. Early works, such as LLaVA-RLHF (Sun et al., 2023) and RLHF-V (Yu et al., 2024a), relied heavily on human-intensive labeling to collect high-quality feedback and correct mistakes in model responses. To alleviate the demand for intensive human labeling, various approaches (Li et al., 2023a; Zhao et al., 2023; Yu et al., 2024b) have been proposed to collect or construct feedback with preferences automatically. For instance, Bai et al. (2023) prompt GPT-4v (OpenAI., 2024b) to collect preference pairs and distill them into a pre-trained VLM. While this method offers ease and convenience, the preference judgment of GPT-4v is not manually verified, posing risks of bias and unreliability. Approaches like HA-DPO (Zhao et al., 2023), POVID (Zhou et al., 2024a), and STIC (Deng et al., 2024) perturb the image and text prompts or inject false statements into model responses to heuristically construct preference pairs. Other techniques, such as RLAIF-V (Yu et al., 2024b) and CSR (Zhou et al., 2024b), employ self-rewarding mechanisms to attain correctness scores or vision-language alignment scores for preferences. In contrast, we propose a fine-grained, verifiable feedback approach that links specific categories of undesired behavior (e.g., false or irrelevant responses) to detailed text spans (e.g., sentences or sub-sentences), which provides more generalizable and reliable automatic feedback for improving learning through feedback.

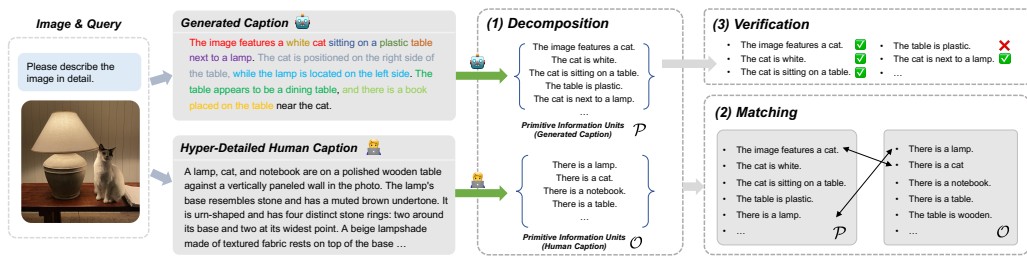

Figure 1: Overview of the proposed DCSCORE for evaluating detailed image captioning. (1) Given the image and prompt, model generated responses and human written responses are decomposed into sets of primitive information units. (2) We match the primitive information units of generated response $\mathcal{P}$ and those of human written response $\mathcal{O}$. (3) Each primitive information unit in $\mathcal{P}$ is verified individually by VLM given the content of images.

## 3   DECAPBENCH: IMAGE CAPTIONING TESTBED FOR MODERN VLMS

Recent open-source VLMs have been significantly improved, narrowing their performance gap compared with GPT-4V on various benchmarks. However, this progress does not always translate into better image captioning abilities. The issue lies in the fact that while current VLMs can generate detailed captions with many fine-grained elements, existing metrics rely on coarse-grained ground-truth captions that overlook these details. Furthermore, traditional automatic evaluation metrics show lower correlation with human evaluations, raising questions about their effectiveness. To address these limitations, we propose DECAPBENCH, a new image captioning evaluation benchmark, along with a novel metric DCSCORE, as illustrated in Figure 1, that better captures the descriptive capabilities of VLMs. Our metric ensures that model rankings align more closely with results from the VLM arena, which is based on diverse, crowd-sourced user votes for image description tasks.

### 3.1   DCSCORE EVALUATION METRIC

Previous image caption evaluation metrics (Papineni et al., 2002; Vedantam et al., 2015; Banerjee & Lavie, 2005; Hessel et al., 2021; Anderson et al., 2016) are designed for short caption evaluation.

When applied to detailed captioning, these metrics suffer from limitations such as low-quality and uninformative annotations, as well as biased captioning patterns, resulting in failures to adequately assess hallucinations and the comprehensiveness of captions generated by VLMs. To address this issue, we propose DCSCORE, a novel metric for detailed image captioning that accounts for both hallucinations and fine-grained comprehensiveness. DCSCORE evaluates the quality of image captions by generating and assessing *primitive information units*, which are the smallest self-sufficient units of information within a caption. This method reduces ambiguity and enhances the transparency of the evaluation process. The evaluation process consists of three steps, described as following.

**Step 1: Decomposition.** The extraction of primitive information units involves splitting the model-generated caption into distinct components, which can be done either manually or by a large language model (LLM). For the ground-truth caption, we use human experts to decompose it into a set of primitive information units, denoted as $\mathcal{O} = \{o_1, o_2, \cdots, o_M\}$, where $M$ is the total number of extracted units. On the other hand, we prompt the LLM to decompose the model-generated caption on a sentence-by-sentence basis into a set $\mathcal{P} = \{p_1, p_2, \cdots, p_N\}$, where $N$ represents the number of units extracted from the model's description. Since image captions can include elements that are not directly descriptive of the image, which may influence the overall quality and style of the caption, it is essential to evaluate these non-descriptive elements as part of the VLMs' captioning capabilities. To differentiate between descriptive and non-descriptive units, we prompt LLMs to perform a binary classification for each unit $p_i \in \mathcal{P}$ during decomposition. Detailed instructions for extracting primitive information units can be found in the Appendix.

**Step 2: Matching.** High-quality model-generated captions should incorporate all key elements from the reference captions without omissions. To evaluate this, we prompt LLMs to assess whether each primitive information unit $p_i \in \mathcal{P}$ from the generated caption is mentioned or can be logically inferred from the reference caption $o_j \in \mathcal{O}$. The matching process is formally computed as $\mathcal{Q} = \mathcal{P} \cap \mathcal{O}$, where $\mathcal{Q}$ is the overlap of primitive information units between the generated and reference captions.

**Step 3: Verification.** To verify the correctness of the primitive information units $p_i$ in the generated captions $\mathcal{P}$, we use modern VLMs. Specifically, we employ GPT-4o (OpenAI., 2024a) to assess the accuracy of each unit by referencing the corresponding image. GPT-4o is prompted to provide a simple "yes" or "no" answer regarding the correctness of each unit, without requiring further explanation, following the approach used by Li et al. (2023b).

After obtaining the model-generated set $\mathcal{P}$, the reference set $\mathcal{O}$, and their overlap $\mathcal{Q}$, we compute both a precision score $s_p$ (non-hallucination) and a recall score $s_r$ (comprehensiveness) as follows:

$$s_p = \frac{|\mathcal{P}_{true}|}{|\mathcal{P}|}, \quad s_r = \frac{|\mathcal{Q}| + |\mathcal{P}_{true} \setminus \mathcal{Q}|}{|\mathcal{O}| + |\mathcal{P}_{true} \setminus \mathcal{Q}|}, \quad (1)$$

where $\mathcal{P}_{true} = \{p_i | p_i \in \mathcal{P}, p_i \text{ is correct}\}$ represents the set of correct units in the set $\mathcal{P}$.

We assess the overall caption quality using the F1 score $s_f$, which balances the precision score $s_p$ and recall score $s_r$. Additionally, we evaluate the descriptive elements of the caption by computing the F1 score $s_f'$ for only the descriptive units. The final assessment score $\mathcal{F}$ is computed as:

$$\mathcal{F} = \frac{1}{2}(s_f + s_f'). \quad (2)$$

## 3.2 DECAPBENCH: A DETAILED IMAGE CAPTIONING EVALUATION BENCHMARK

**Dataset.** We consider the recently released ImageInWords dataset (Garg et al., 2024), and leverage 400 high-quality, human-curated public image detailed captions from as the ground-truth captioning. Compared with ImageInWords, traditional caption datasets such as COCO (Sidorov et al., 2020; Lin et al., 2014; Agrawal et al., 2019) often contains short, coarse-grained captions, and lack detailed information, making them inadequate for measuring the correctness and comprehensiveness of the models' generated detailed captions. In contrast, ImageInWords considers a human-in-the-loop framework produces hyper-detailed and hallucination-free image descriptions, by combining human annotators and seeded machine generations. Consequently, we constructed DECAPBENCH,

| Metric | PCC ($\rho$) ↑ | $1 - R^2$ ↓ | Kd $\tau$ ↑ | Sp $\tau$ ↑ |
|---|---|---|---|---|
| *Rule-Based Evaluation* | | | | |
| BLEU-4 (Papineni et al., 2002) | 0.3439 | 62.78 | 0.2693 | 0.2931 |
| ROUGE (Lin, 2004) | 0.2509 | 156.05 | 0.1886 | 0.1893 |
| METEOR (Banerjee & Lavie, 2005) | 0.3593 | 111.95 | 0.2417 | 0.2536 |
| CIDEr (Vedantam et al., 2015) | 0.0522 | 3.3e7 | 0.0635 | 0.0601 |
| *Model-Based Evaluation* | | | | |
| SPICE (Anderson et al., 2016) | 0.2218 | 156.11 | 0.1731 | 0.1907 |
| CLIP-Score (Hessel et al., 2021) | 0.2183 | 26.04 | 0.1724 | 0.1480 |
| PAC-Score (Sarto et al., 2023) | 0.1525 | 20.93 | 0.1117 | 0.1260 |
| CAPTURE (Dong et al., 2024) | 0.3521 | 7.62 | 0.2801 | 0.3449 |
| CLAIR (Chan et al., 2023) | 0.3815 | 1.98 | 0.3847 | 0.4552 |
| FLEUR (Lee et al., 2024) | 0.4230 | 3.01 | 0.4246 | 0.5325 |
| GPT4-Eval (Liu et al., 2024b) | 0.3976 | 2.95 | 0.3447 | 0.3866 |
| Faithscore (Jing et al., 2023) | 0.1937 | 3.22 | 0.1626 | 0.1115 |
| RLAIF-V (Yu et al., 2024b) | 0.3547 | 5.32 | 0.2774 | 0.2544 |
| **DCSCORE** | **0.6605** | **1.54** | **0.5328** | **0.6166** |

Table 1: Correlation of image captioning evaluation metrics and human judgements. All p-values $< 0.001$. The bold number indicates the highest human consistency among all caption metrics.

by applying the proposed DCSCORE evaluation metric to the ImageInWords images and their corresponding hyper-detailed image captions.

**Human consistency of DCSCORE.** To demonstrate consistency with human expert judgments, we randomly selected 500 captions generated by different models and employed X experienced annotators to rate each caption. We then computed the statistical metrics to compare the proposed DCSCORE with human ratings, including the Pearson correlation coefficient (PCC) $\rho$, coefficient of determination $R^2$, Kendall's $\tau$ (Kd $\tau$) and Sample-wise $\tau$ (Sp $\tau$). The correlation statistics, as presented in Figure 2 (Left), highlight the significant improvements brought by our proposed metric, DCSCORE. Compared to the state-of-the-art, DCSCORE enhances PCC $\rho$ by 0.2375 and boosts Kendall $\tau$ by 0.1082. These advancements suggest that our metric achieves superior linear correlation and pairwise ranking accuracy with human judgments. Hence, DCSCORE holds great potential for optimizing detailed captions produced by VLMs.

High-quality and hyper-detailed image descriptions are crucial for evaluating model-generated captions, as they closely mirror the content of the image. To investigate this, we assess the impact of varying quality of ground-truth descriptions on our proposed DCSCORE. As shown in Figure 2 (Left), descriptions with finer granularity achieve higher consistency with human judgments compared to COCO-style concise captions. Specifically, detailed captions annotated by either humans or GPT-4o (OpenAI, 2024a) demonstrate a superior alignment with human evaluators, highlighting the importance of granularity in image description for more reliable and accurate evaluation.

| Source of Captions | PCC ($\rho$) ↑ | $1 - R^2$ ↓ | Kd $\tau$ ↑ | Sp $\tau$ ↑ |
|---|---|---|---|---|
| COCO-Style | 0.5468 | 14.10 | 0.4375 | 0.5093 |
| Instruct-BLIP | 0.6062 | 5.50 | 0.4745 | 0.5620 |
| GPT-4o | 0.6497 | 2.03 | 0.5194 | 0.5745 |
| Human Annotated | **0.6605** | **1.54** | **0.5328** | **0.6166** |

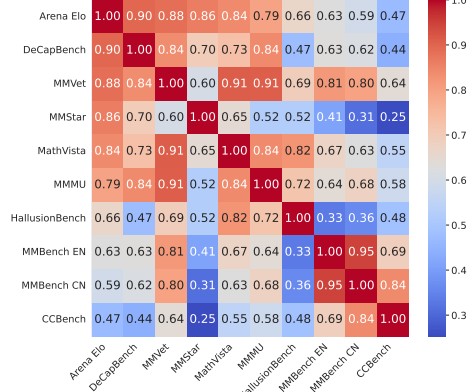

Figure 2: (Left) Comparison of four sources for ground-truth captions in terms of correlation between DCSCORE and human judgments. All p-values are less than $0.001$. (Right) DECAPBENCH achieves the highest correlation with Arena Elo, with a Spearman's correlation of 0.90 among different VLM benchmarks.

**Human consistency of DECAPBENCH.** To further study the consistency between the proposed DECAPBENCH and human judgement in the wild, we select the subset of image description from the VLM arena, and compute the ranking correlation. Note that VLM arena is a public VLM evaluation platform, where two model responses for the same task prompt are voted by humans to reflect their preferences. Specifically, we compute human preferences using Elo ratings, derived from over 1,000 pairwise comparisons with around 800 images across 13 different VLMs on image captioning tasks.

In Figure 2 (Right), we visualize the Spearman correlation heatmap among various automatically evaluated multi-modal benchmarks (Chen et al., 2024a; Liu et al., 2023; Yue et al., 2024; Kembhavi et al., 2016) and human-voted preference benchmarks (Lu et al., 2024). From the figure, we observe that DECAPBENCH achieves the highest correlation with Arena Elo at 0.90, indicating a high level of alignment with human preferences and a strong consistency in ranking. This high correlation demonstrates the effectiveness of DECAPBENCH in capturing the nuances of human judgment, making it a reliable benchmark for evaluating the image captioning capabilities of VLMs.

Compared with existing multimodal benchmark, the proposed DECAPBENCH is unique in its dedication to the task of detailed captioning, verified by the highest correlation with Arena captoin subset. Note that MMVet (Yu et al., 2023) evaluates the models' ability to solve complex vision-language tasks. MMMU (Yue et al., 2024) and MathVista (Lu et al., 2023) assess subject knowledge and mathematical reasoning in visual contexts, respectively, while HallusionBench focuses on understanding visually misleading figures. The MMBench-series (Liu et al., 2023) (e.g., MMBench-EN, MMBench-CN, and CCBench) concentrates on fine-grained perception and reasoning tasks using multiple-choice questions. Additionally, MMStar (Chen et al., 2024a) corrects the misjudgments of actual multi-modal performance.

# 4 LEARNING FROM FINE-GRAINED FEEDBACK

## 4.1 FINE-GRAINED FEEDBACK COLLECTION

The feedback collected for preference learning consists of comparison pairs, where each pair includes a preferred response and a less preferred response to the same input. The model learns from this preference data to distinguish differences among its own generated candidate responses. To gather these candidate responses, we generate multiple outputs for given images and prompts using nucleus sampling (Holtzman et al., 2019), varying the random seed to ensure diversity. By learning to rank these candidate responses based on the preference data, the model becomes capable of assessing the quality of its outputs and deriving appropriate signals for preference optimization.

However, judging the quality of different responses is complex, even for experienced human annotators (Sun et al., 2023), due to the semantic intricacies involved. Previous methods (Zhou et al., 2024a; Zhao et al., 2023) attempted to address this by manually modifying responses and injecting noise to create negative samples. However, these approaches suffer from poor generalization because of implicit patterns in the data. In contrast, by adapting the concept of primitive information units and step-by-step verification (Lightman et al., 2023), we propose FEEDQUILL for feedback collection, which leverages modern VLMs to generate fine-grained feedback in the following three steps:

- *Decomposition*. We prompt an LLM to decompose the response into a set of $N$ primitive information units $\{p_i\}_{i=1}^N$ on a sentence-by-sentence basis, rewriting them into self-sufficient and verifiable statements.
- *Scoring*. We use several powerful VLMs (Chen et al., 2024b; Liu et al., 2024a) to verify these rewritten statements using the prompt: `"{STATEMENT} Is the statement correct? Please only answer 'yes' or 'no'"`. To increase confidence in our judgments, we ensemble the results from multiple open-source VLMs for verification.
- *Preference*. After obtaining the verification results for each primitive information unit, we calculate the preference score $c_p$ as the fraction of correct units: $c_p = \frac{1}{N}\sum_{i=1}^N \mathbb{1}\{p_i = 1\}$, where a higher score indicates fewer hallucinations in the response. Given the scores of each response, we construct a preference dataset $\mathcal{D} = (x_i, y_i^+, y_i^-)$ by treating the response with the higher score as the preferred response $y_i^+$ and the one with the lower score as the non-preferred response $y_i^-$.

As discussed in Zhu et al. (2023), responses with fewer hallucinations are often inherently less helpful. Specifically, models are more likely to hallucinate when producing longer responses compared to

shorter ones. To address this issue, we construct a preference dataset $\mathcal{D}_r$ using the number of primitive information units as the preference score $c_r$. A response with a higher score $c_r$ — indicating more primitive information units — is considered more preferable. This approach encourages the model to generate responses that are not only accurate but also rich in helpful and detailed information.

## 4.2 PREFERENCE OPTIMIZATION

Preference optimization (Ouyang et al., 2022; Rafailov et al., 2024) has shown promise in fine-tuning language models and aligning their behavior with desired outcomes. Specially, we train the reward model $r_\phi$ with the preference set $\mathcal{D}$ and $\mathcal{D}_r$ respectively, with the a pairwise comparison loss (Ouyang et al., 2022) as $\mathcal{L}_{RM} = -\mathbb{E}_{(x,y^+,y^-)\sim\mathcal{D}} \left[\log\left(\sigma(r_\phi(x,y^+) - r_\phi(x,y^-))\right)\right]$, where $\sigma(\cdot)$ is the sigmoid function and $r_\phi(\cdot,\cdot)$ is the output score of the reward model. To mitigate biased preferences, such as unhelpful responses, we opt against using a single scalar reward to represent response quality. Instead, we leverage rewards derived from multiple reward models, each contributing to distinct behaviors like hallucination ($c_p$) and richness ($c_r$). To optimize these preferences, we utilize proximal policy optimization (PPO) (Schulman et al., 2017), a widely adopted reinforcement learning algorithm. To fully exploit the characteristics of preferences related to hallucination and comprehensiveness, we select captioning as the optimization task. For additional details, please refer to the Appendix.

## 5 EXPERIMENTS

### 5.1 EXPERIMENTAL SETUP

**Model.** We conduct our experiments based on a series of LLaVA models (Liu et al., 2024b) with different sizes and capabilities. We initialize both the policy model and reward model with same parameters as well as same size for validating the effectiveness of our proposed method. For the main results, we report the performance of our model FEEDQUILL-7B trained on LLaVA-Onevision-7B, one of the most capable models in the < 10B size category.

**Training Dataset for PPO.** The PPO is performed with the detailed captioning task. To ensure the model learns robust generalization capabilities, diversity in image distributions is crucial. Therefore, we randomly sample images from a wide range of datasets, including MSCOCO (Lin et al., 2014), OpenImages (Kuznetsova et al., 2020), and ShareGPT4V (Chen et al., 2023). Additionally, to maintain diversity of instructions during training, we prompt GPT-4o (OpenAI., 2024a) to generate a variety of caption prompts, and provide in Appendix.

### 5.2 ABLATIONS

**Preference Data for Reward Model.** To assess the ability of various preference data to generalize, we trained multiple reward models using the same SFT model. For evaluation, we randomly sampled portions of the preference data that were held out. The findings, presented in Table 2, reveal that our model achieved the highest accuracy across diverse preference datasets. Notably, with the same scale of training data, our reward model outperformed the human-labeled dataset RLHF-V by 9.9% in accuracy. It also surpassed the RLAIF-V dataset, which, despite having over 80k training samples, was outperformed by our model that utilized a smaller data size. Additionally, we observed that increasing the amount of training data led to an improvement in average accuracy from 71.3% to 75.2%, highlighting the scalability of our approach.

**Preference Data for Preference Optimization.** We delve into how varying types of preference data impact preference optimization. Using LLaVA-1.5-7B as our baseline model, we trained it with a variety of preference datasets. The performance of these models was then assessed through a range of downstream benchmarks in a zero-shot context. As showcased in Table 3, our approach not only excels in captioning performance but also substantially cuts down on hallucinations, achieving a notable 0.75 improvement on mmHal-V compared to the baseline.

**Data Size.** We scale up the training set of the reward model, and investigate the correlation between downstream performance through preference optimization. We evaluate different checkpoints ranging

| Train Data | Held-Out Eval Dataset | | | | | | |
|---|---|---|---|---|---|---|---|
| | HA-DPO | RLHF-V | POVID | CSR | RLAIF-V | STIC | Average |
| HA-DPO (Zhao et al., 2023) | 93.5 | 81.1 | 23.7 | 53.5 | 51.0 | 42.0 | 57.5 |
| RLHF-V (Yu et al., 2024a) | 82.0 | 94.7 | 30.7 | 44.2 | 48.7 | 67.8 | 61.4 |
| POVID (Zhou et al., 2024a) | 32.5 | 30.6 | 99.5 | 59.4 | 52.5 | 59.5 | 55.7 |
| CSR (Zhou et al., 2024b) | 62.5 | 51.8 | 60.3 | 87.5 | 51.8 | 23.6 | 56.3 |
| RLAIF-V (Yu et al., 2024b) | 69.5 | 49.5 | 77.6 | 55.5 | 68.1 | 66.8 | 64.5 |
| STIC (Deng et al., 2024) | 48.0 | 59.7 | 26.8 | 43.3 | 50.1 | 99.9 | 54.6 |
| FEEDQUILL* | 78.0 | 64.1 | 87.4 | 59.7 | 64.7 | 74.1 | **71.3** |
| FEEDQUILL | 76.5 | 71.9 | 93.2 | 55.2 | 69.4 | 84.9 | **75.2** |

Table 2: Reward model zero-shot accuracy on the held-out validation set trained with different preference data on LLaVA-1.5-7B. * indicates that we only utilize 10k preference data to match the size of other training set.

| Method | MMBench ↑ | VizWiz ↑ | MMStar ↑ | WildVision ↑ | LLaVA-W ↑ | DECAPBENCH ↑ | mmHal-V ↑ | $CHAIR_S$ ↓ | $CHAIR_I$ ↓ |
|---|---|---|---|---|---|---|---|---|---|
| LLaVA-1.5 | 64.8 | 50.0 | 33.1 | 14.48 | 65.3 | 24.50 | 1.85 | 47.8 | 25.3 |
| w/ HA-DPO | 64.3 | 54.1 | 33.5 | 15.17 | 65.1 | 22.45 | 2.12 | 49.3 | 25.5 |
| w/ POVID | 64.7 | 47.9 | 35.4 | 13.25 | 71.5 | 23.54 | 1.90 | 31.8 | 5.4 |
| w/ CSR | 64.2 | 52.8 | 33.8 | 13.85 | 70.3 | 23.70 | 2.12 | 15.7 | 7.9 |
| w/ RLAIF-V | 62.7 | 50.9 | 34.7 | 15.65 | 76.0 | 28.21 | 2.59 | 8.5 | 4.3 |
| w/ FEEDQUILL | **66.3** | **55.2** | **35.8** | **19.68** | **76.0** | **34.52** | **2.60** | **5.1** | **2.6** |

Table 3: The performance of different preference data on LLaVA-1.5-7B across different benchmarks.

from 5,000 to 200,000 training samples, using models of sizes 7B and 13B. The results are illustrated in Figure 3. As the size of the preference data increased, the performance of mmHal-V improves from 2.05 to 2.6. Similarly, MMStar, which focuses on image understanding, shows a consistent increase from 34.7 to 35.8, yielding a 1.1 point lift. This demonstrates that as the size of preference data for the reward model grows, the model's performance consistently improves since the better reward model provides more accurate signals for preference optimization.

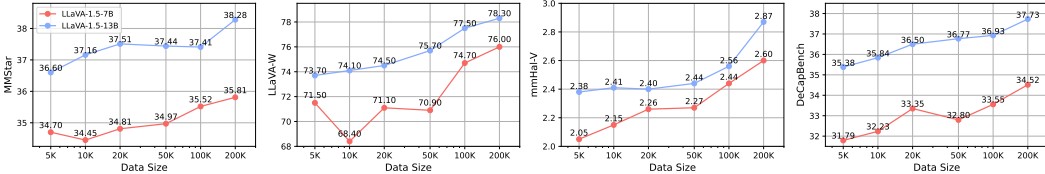

Figure 3: Impact of the preference dataset size in terms of downstream performance.

**Source of Responses.** We explore the effect of the source of model responses on preference data, based on the hypothesis that improvements might arise from the model's ability to generalize across varying sources. To test this hypothesis, we use LLaVA-1.5-13B as the base model and examine responses sampled either from the same model or from other models such as LLaVA-1.5-7B, LLaVA-1.6-7B, and LLaVA-1.6-13B. Furthermore, we assess the impact of combining responses from these different sources. The results of these experiments are summarized in Table 4. We observe that integrating responses generated by the same model only leads to a significant performance boost compared to the baseline model. Conversely, integrating responses from different models only leads to larger performance gains on DECAPBENCH by providing diverse responses, while smaller gains on other benchmarks. When combining responses from both sources, the model achieves superior performance, surpassing the use of either source alone. Specifically, this combination results in an improvement of 13.0 points on LLaVA-W and 13.23 points on DECAPBENCH compared to baseline.

**Source of Rewards.** Table 5 provides a comparative analysis of incorporating the preference score for the number of primitive information units ($c_r$) alongside the preference score for the proportion of correct units ($c_p$). Each preference score is obtained separately from different reward models, summed to a final reward in PPO training procedure. We specifically evaluate our method against three distinct variants: (1) the base model without any preference optimization (Base); (2) a model optimized solely with the $c_p$ score (Only $c_p$); and (3) a model optimized exclusively with the $c_r$ score (Only $c_r$). This comparison enables a thorough examination of the impact of each optimization strategy on model performance. Notably, models trained with the $c_p$ score consistently enhance performance on both LLaVA-W and DECAPBENCH. Conversely, models trained with the $c_r$ score

| Source of Response | | MMStar | LLaVA-W | mmHal-V | DECAPBENCH |
|---|---|---|---|---|---|
| Same Model | Other Models | | | | |
| | | 33.1 | 65.3 | 1.85 | 24.50 |
| ✓ | | 37.6 | 75.1 | 2.74 | 26.32 |
| | ✓ | 38.0 | 71.5 | 2.53 | 34.84 |
| ✓ | ✓ | **38.3** | **78.3** | **2.83** | **37.73** |

Table 4: Comparison of performance by varying sources of preference data.

| Method | LLaVA-1.5-7B | | LLaVA-1.5-13B | |
|---|---|---|---|---|
| | LLaVA-W | DECAPBENCH | LLaVA-W | DECAPBENCH |
| Base | 65.3 | 24.50 | 72.8 | 25.55 |
| Only $c_p$ | 67.3 | 25.21 | 74.3 | 26.23 |
| Only $c_r$ | 46.2 | 10.03 | 56.9 | 15.11 |
| $c_p + c_r$ | **76.0** | **34.52** | **78.3** | **37.73** |

Table 5: Ablation of using different reward scores during preference optimization.

yield poorer results on both datasets due to the absence of a precision constraint. Furthermore, when both $c_p$ and $c_r$ are incorporated, our method exhibits significant improvements, notably a 10.7% increase on LLaVA-1.5-7B and a 5.5% boost on LLaVA-1.5-13B.

| Method | Comprehensive Benchmark | | | | Visual Hallucination | Visual Chat and Captioning | | |
|---|---|---|---|---|---|---|---|---|
| | MMBench | MMStar | VizWiz | SciQA$^I$ | mmHal-V | LLaVA-W | WildVision | DECAPBENCH |
| LLaVA-1.5-7B | 64.8 | 33.1 | 50.0 | 66.8 | 1.85 | 65.3 | 14.48 | 24.50 |
| + FEEDQUILL | 66.3 (+1.7) | 35.8 (+2.7) | 55.2 (+5.2) | 68.9 (+2.1) | 2.60 (+0.75) | 76.0 (+10.7) | 17.68 (+3.20) | 34.52 (+10.02) |
| LLaVA-1.5-13B | 68.7 | 34.3 | 53.6 | 71.6 | 2.33 | 72.8 | 16.17 | 25.55 |
| + FEEDQUILL | 69.2 (+0.5) | 38.3 (+4.0) | 56.8 (+3.2) | 73.4 (+1.8) | 2.83 (+5.00) | 78.3 (+5.5) | 18.15 (+1.98) | 37.73 (+12.18) |
| LLaVA-1.6-7B | 67.1 | 37.6 | 57.6 | 70.2 | 2.58 | 79.8 | 26.15 | 35.74 |
| + FEEDQUILL | 67.9 (+0.8) | 38.6 (+1.0) | 63.4 (+5.8) | 70.3 (+0.1) | 2.93 (+0.35) | 82.4 (+2.6) | 44.16 (+18.01) | 52.69 (+16.95) |
| LLaVA-1.6-13B | 69.3 | 40.4 | 60.5 | 73.6 | 2.95 | 85.2 | 33.69 | 36.28 |
| + FEEDQUILL | 69.9 (+0.6) | 41.1 (+0.7) | 66.7 (+6.2) | 73.5 (+0.1) | 3.76 (+0.81) | 87.1 (+1.9) | 49.69 (+16.00) | 53.26 (+16.98) |
| LLaVA-Onevision-7B | 80.8 | 61.7 | 60.0 | 96.0 | 2.94 | 90.7 | 54.50 | 43.49 |
| + FEEDQUILL | 80.5 (-0.3) | 62.4 (+0.7) | 60.4 (+0.4) | 95.9 (-0.1) | 3.10 (+0.16) | 100.5 (+9.8) | 59.60 (+5.10) | 55.65 (+12.16) |

Table 6: Performance of FEEDQUILL with various VLM models on downstream tasks.

**Compatibility Analysis.** To validate the applicability of FEEDQUILL across various VLMs, we conduct experiments on various models. The summarized results in Table 6 reveal that FEEDQUILL is effective regardless of model size, consistently enhancing performance on downstream tasks such as MMBench, mmHal-V, and DECAPBENCH. This underscores the robust generalization capability of our proposed FEEDQUILL. Notably, LLaVA-1.6-13B trained with FEEDQUILL exhibits large improvement on mmHal-V, increasing from 2.95 to 3.76. Simultaneously, it significantly boosts performance on WildVision and DECAPBENCH, with gains of +16.0% and +16.98%, respectively.

## 5.3 MAIN RESULTS

| Model | AI2D | ChartQA | MMBench | SEEDBench | MME | MMMU | MMVet | MMStar | SciQA | LLaVA-W | WildVision | DECAPBENCH |
|---|---|---|---|---|---|---|---|---|---|---|---|---|
| *Proprietary Model* | | | | | | | | | | | | |
| Claude-3.5-Sonnet | 94.7 | 90.8 | 78.5 | - | -/- | 68.3 | 75.4 | 60.2 | 80.5 | 102.9 | 50.00 | 52.37 |
| Gemini-1.5-Pro | 94.4 | 87.2 | 73.9 | - | -/- | 62.2 | 64.0 | 58.7 | - | - | 35.45 | 46.34 |
| GPT-4V | 78.2 | 78.5* | 79.8 | 49.9 | 1409/517 | 56.8 | 57.1 | 75.7 | 75.7 | 98.0 | 80.01 | 48.52 |
| GPT-4o | 94.2 | 85.7 | 80.5 | 76.2 | -/- | 69.1 | 76.2 | 59.8 | 83.5 | 106.1 | 89.41 | 53.44 |
| *Open-Source Model* | | | | | | | | | | | | |
| Cambrian-34B | 79.7 | 73.8 | 81.4 | - | -/- | 49.7 | 53.2 | 85.6 | 67.8 | - | - | 35.12 |
| VILA-40B | - | - | 82.4 | 75.8 | 1762 | 51.9 | 51.2 | 54.2 | - | - | - | 38.02 |
| XComposer-2.5-7B | 81.5 | 82.2 | 82.2 | 75.4 | 2229 | 42.9 | 51.7 | 59.9 | - | 78.1 | - | 29.60 |
| InternVL-2-8B | 83.8 | 83.3 | 81.7 | 76.0 | 2210 | 49.3 | 60.0 | 59.4 | 97.0 | 84.5 | - | 45.55 |
| InternVL-2-26B | 84.5 | 84.9 | 83.4 | 76.8 | 2260 | 48.3 | 65.4 | 60.4 | 97.5 | 99.6 | - | 49.59 |
| LLaVA-Onevision-7B | 81.4 | 80.0 | 80.8 | 75.4 | 1580/418 | 48.8 | 57.5 | 61.7 | 96.0 | 90.7 | 54.50 | 43.49 |
| FEEDQUILL-7B | 81.3 | 80.3 | 80.5 | 75.8 | 1515/450 | 47.9 | 59.3 | 62.4 | 95.9 | **100.5** | 59.60 | **55.65** |

Table 7: Main experimental results of our method and other open-sourced state-of-the-art VLMs. *GPT-4V reports 4-shot results on ChartQA. All results are presented in the 0-shot setting.

We evaluate FEEDQUILL-7B across a variety of multi-modal large language model benchmarks, including AI2D (Kembhavi et al., 2016), ChartQA (Masry et al., 2022), MMBench (Liu et al., 2023), SEEDBench (Li et al., 2024b), MME (Fu et al., 2023), MMMU (Yue et al., 2024), MMVet (Yu et al., 2023), MMStar (Chen et al., 2024a), ScienceQA (Lu et al., 2022), LLaVA-W Liu et al. (2024b), WildVision (Lu et al., 2024), and DECAPBENCH. These datasets are specifically designed to measure various capabilities of VLMs, including document understanding, question answering, visual chatting, visual perception, and detailed image captioning. Table 7 presents a comparative analysis of FEEDQUILL-7B against state-of-the-art VLMs, encompassing both proprietary and open-source

models including Claude-3.5-Sonnet (Anthropic., 2024), Gemini-1.5-Pro (Team et al., 2023), GPT-4v (OpenAI., 2024b), GPT-4o (OpenAI., 2024a), Cambrian-34B (Tong et al., 2024), VILA-40B (Lin et al., 2024), XComposer-2.5-7B (Zhang et al., 2024), and InternVL-2-8B/26B (Chen et al., 2024b).

FEEDQUILL-7B achieves state-of-the-art performance in detailed image captioning, surpassing GPT-4o by 2.21 points. Remarkably, it also outperforms GPT-4v on LLaVA-W, showing strong capability in visual chatting. Despite being trained solely on the captioning task, our model maintains its strong performance while achieving a 1.8-point improvement on MMVet and a 0.7-point increase on MMStar compared to LLaVA-Onevision-7B. Additionally, it retains most of its capabilities after preference optimization – a feat that many aligned models, such as BHDS (Amirloo et al., 2024), CSR (Zhou et al., 2024b), and RLAIF-V (Yu et al., 2024b), fail to accomplish.

## 5.4 CASE STUDY

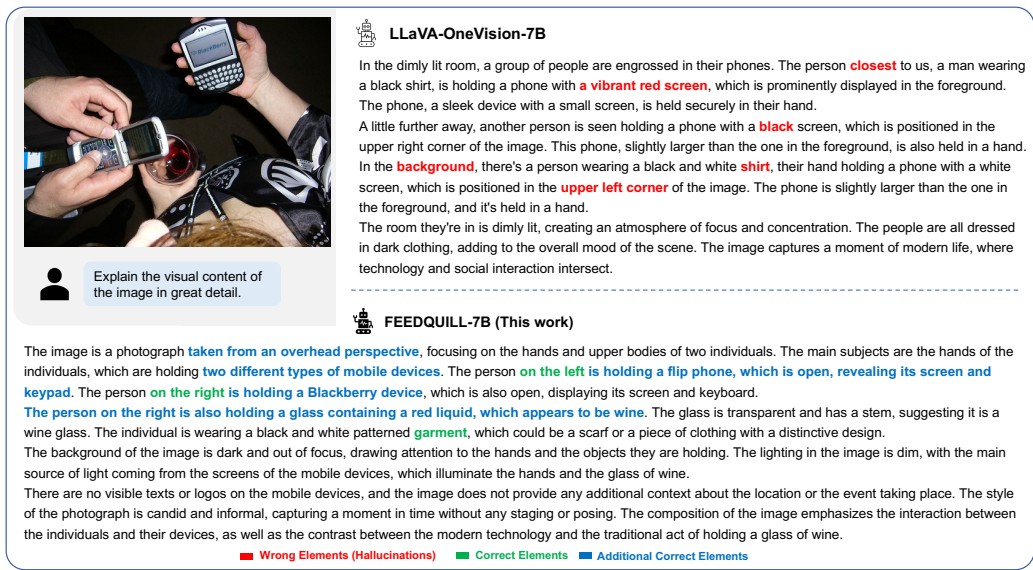

Figure 4: Qualitative results of FEEDQUILL-7B compared with LLaVA-Onevision-7B (Li et al., 2024a) in terms of image captioning.

We provide qualitative results of LLaVA-Onevision-7B and FEEDQUILL-7B in Figure 4 to illustrate the effectiveness of our proposed method. In the example above, LLaVA-Onevision-7B incorrectly identifies the red wine in the glasses as a vibrant screen. In contrast, our model correctly identifies it as a red liquid with fewer instances of hallucination. Additionally, while LLaVA-Onevision-7B generically names both phone as "cell phone", FEEDQUILL-7B specifically identifies them as a Blackberry device and a flip phone, showcasing its strong fine-grained captioning capabilities. We refer readers to the Appendix for more qualitative results.

## 6 CONCLUSION

We have described a novel metric, DCSCORE, designed to evaluate both hallucination and comprehensiveness, the two critical challenges in detailed image captioning. Empirical validations show that DCSCORE achieves the highest consistency with human judgments, underscoring its reliability. Additionally, we present a new detailed caption benchmark, DECAPBENCH, specifically for assessing the captioning capabilities of modern VLMs. Our results demonstrate that the correlation of DECAPBENCH with human judgment surpasses that of any other public benchmark in description tasks. Furthermore, we propose an effective fine-grained feedback collection method, FEEDQUILL, which decomposes responses into primitive information units for individual verification and subsequently learns an improved model through preference optimization. Comprehensive experiments reveal that FEEDQUILL is applicable across various models, achieving superior image captioning performance while reducing hallucinations, and setting new state-of-the-art. We believe that both DECAPBENCH and FEEDQUILL will serve as invaluable foundations for advancements in detailed image captioning and preference optimization.

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

# A APPENDIX

## A.1 DISCUSSION

### A.1.1 RELATED WORKS

|  | Faithscore | RLAIF-V | Ours |
|---|---|---|---|
| Descriptive/Non-Descriptive | ✓ / ✗ | ✓ / ✗ | ✓ / ✓ |
| Response Evaluation Coverage | Full | Partial | Full |
| Hallucination | ✓ | ✓ | ✓ |
| Comprehensiveness | ✗ | ✗ | ✓ |
| Decomposition Method | Rewrite | Question-Answer Pairs | Rewrite |
| For Evaluation | ✓ | ✗ | ✓ |
| For Preference Learning | ✗ | ✓ | ✓ |
| Human Correlation (PCC $\rho$) | 0.1937 | 0.3547 | **0.6605** |
| Human Correlation (Kd $\tau$) | 0.1626 | 0.2274 | **0.5328** |
| Human Correlation (Sp $\tau$) | 0.1115 | 0.2544 | **0.6166** |

Table 8: The comparison among related works.

We have compared Faithscore (Jing et al., 2023) and RLAIF-V (Yu et al., 2024b), two metrics built on a similar conceptual foundation, and the distinctions are detailed in Table 8. Below, we summarize these differences to highlight our main contributions:

- **Granularity**: While Faithscore and RLAIF-V evaluate the descriptive aspects of responses, they neglect the non-descriptive elements, which are crucial for caption quality. For example, incorrect assertions about the image's context and inferences can significantly impair understanding. However, in the realm of detailed image captioning, comprehensiveness is equally critical, as shorter captions may indeed exhibit lower hallucination rates but often suffer from a lack of informative value. Our approach uniquely addresses this by simultaneously considering both descriptive and non-descriptive components.

- **Decomposition Method**: Like Faithscore, our method decomposes responses sentence-by-sentence, yet it also includes non-descriptive elements. RLAIF-V, on the other hand, generates question-answer pairs for verification, potentially omitting crucial details.

- **Score Generation**: Faithscore rates the proportion of correct statements, while RLAIF-V counts incorrect statements, which may encourage the model to avoid making any assertions or to state irrelevant but correct information. Conversely, our approach evaluates both the proportion of correct statements for hallucination and the number of valid statements for comprehensiveness.

- **Application**: Our method, designed for detailed image captioning, serves both evaluation and preference learning within a unified framework. Faithscore and RLAIF-V are limited to evaluating or optimizing hallucinations independently.

- **Human Consistency**: Our approach demonstrates the highest correlation with human judgment across various aspects, as shown in the table, validating its effectiveness for detailed image captioning.

In essence, our method introduces a more granular, comprehensive, and human-aligned evaluation framework that surpasses existing methods for detailed image captioning.

## A.2 ADDITIONAL EXPERIMENTS

We investigated the influence of omission elements and non-descriptive elements in DCSCORE on its alignment with human judgment in Table 9 and Table 10 respectively. The results show that including omission elements and non-descriptive elements during detailed image caption evaluation achieves a higher correlation with human judgment. This improvement occurs because non-descriptive elements, such as background details and inferred information, provide additional context that leads to a more comprehensive understanding of the image content. Consequently, by including these elements,

| Omission in GT | PCC ($\rho$) ↑ | $1 - R^2$ ↓ | Kd $\tau$ ↑ | Sp $\tau$ ↑ |
|---|---|---|---|---|
| | 0.6151 | **0.72** | 0.5111 | 0.5916 |
| ✓ | **0.6605** | 1.54 | **0.5328** | **0.6166** |

Table 9: Correlation of DCScore and human judgement in terms of considering omission in ground-truth annotation.

| Non-Descriptive | PCC ($\rho$) ↑ | $1 - R^2$ ↓ | Kd $\tau$ ↑ | Sp $\tau$ ↑ |
|---|---|---|---|---|
| | 0.6213 | 2.77 | 0.5048 | 0.5985 |
| ✓ | **0.6605** | **1.54** | **0.5328** | **0.6166** |

Table 10: Correlation of DCScore and human judgement in terms of considering non-descriptive elements in the captions.

DCScore captures subtle nuances and implicit information critical for fully understanding the image, thus more closely aligning with human judgment.

### A.2.1 CASE STUDY

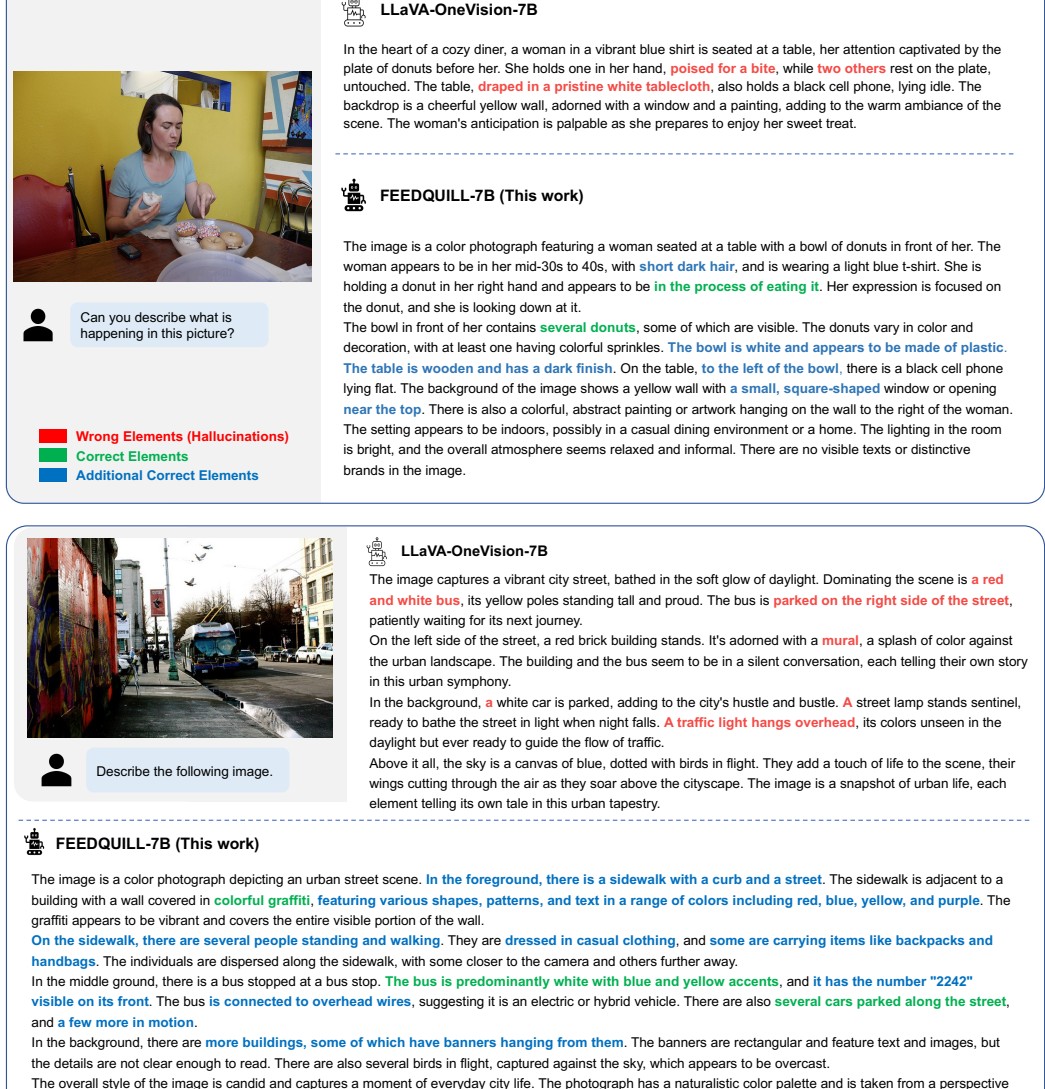

Figure 5: Qualitative results of FEEDQUILL-7B compared with LLaVA-Onevision-7B (Li et al., 2024a) in terms of image captioning.(1)

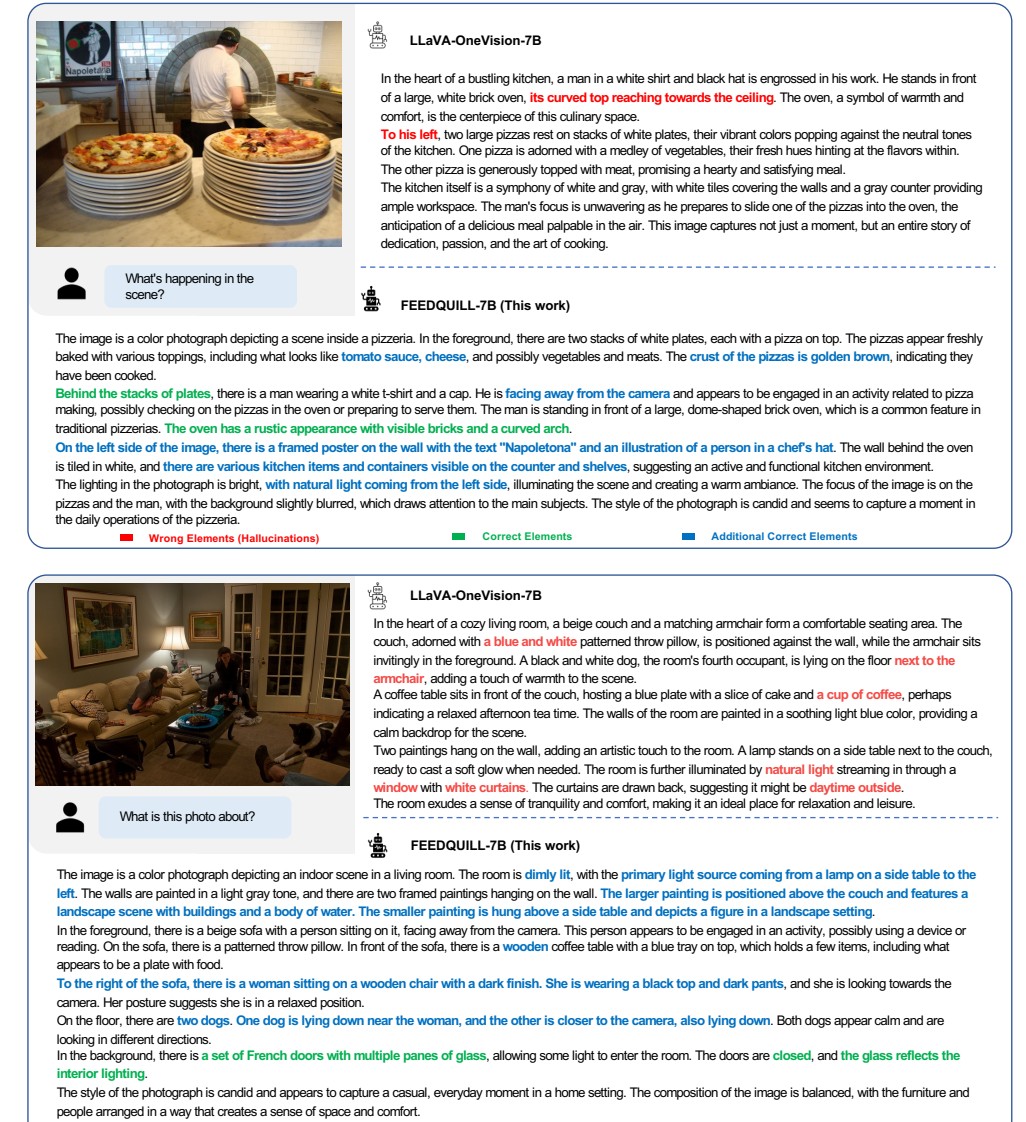

Figure 6: Qualitative results of FEEDQUILL-7B compared with LLaVA-Onevision-7B (Li et al., 2024a) in terms of image captioning.(2)

As instances in Figure 5 and Figure 6 indicates, FEEDQUILL-7B not only significantly reduces hallucinations, but also remarkably improves the granularity and richness of descriptions compared with LLaVA-Onevision-7B (Li et al., 2024a), which is the initial model of FEEDQUILL-7B. From these case we can see the preference score of precision ($c_p$) and the preference of recall ($c_r$) jointly determine the direction of preference optimization in FEEDQUILL, leading the descriptions of the images more precise and more comprehensive. Additionally, we present qualitative results of FEEDQUILL-7B and GPT4o (OpenAI, 2024a) in Figure 7. In these cases GPT4o still introduce hallucinations while FEEDQUILL-7B describe them precisely. From these examples we can get an intuitive understanding of the superior image captioning performance FEEDQUILL-7B achieves.

### A.2.2 THE PERFORMANCE OF VLMS ON DECAPBENCH

We present the performance of various current VLMs on DECAPBENCH in Table 11. As shown, the performance in detailed image captioning consistently improves with an increase in model size. For

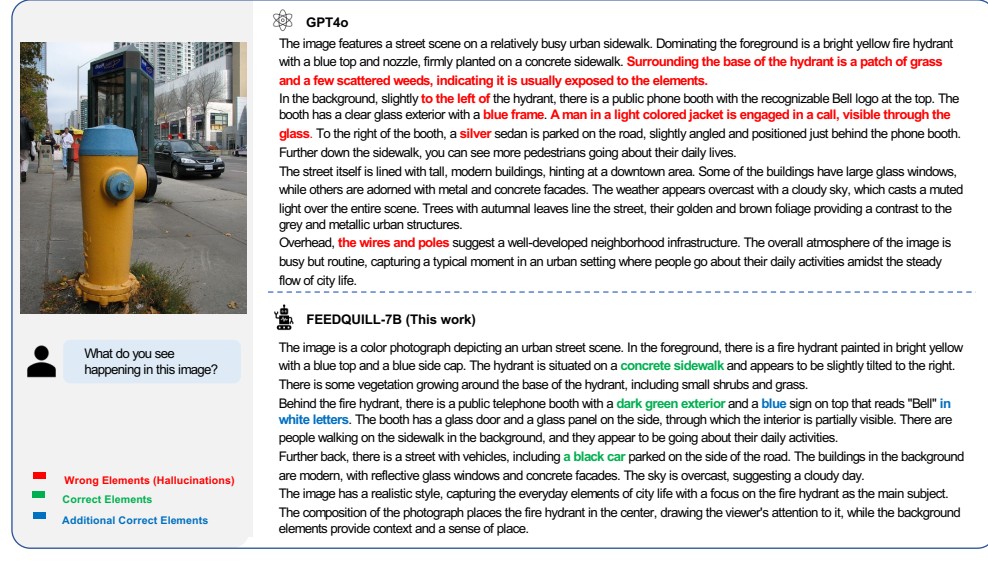

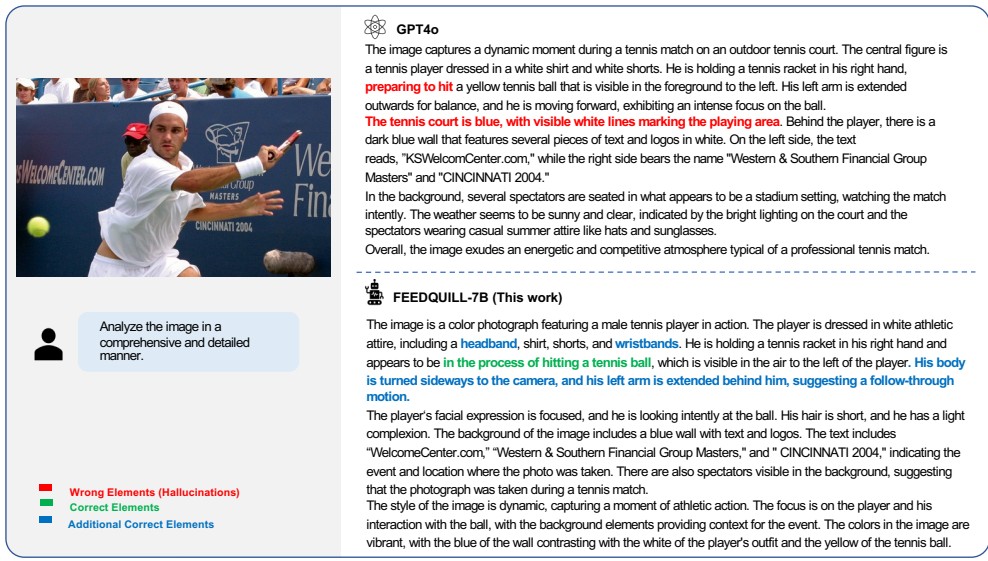

Figure 7: Qualitative results of FEEDQUILL-7B compared with GPT4o (OpenAI., 2024a) in terms of image captioning.

instance, notable improvements are observed in the InternVL-2 series (8/26/40B) (Chen et al., 2024b) and the LLaVA-series (7/13/34B) (Liu et al., 2024a).

## A.3   IMPLEMENTATION

### A.3.1   TRAINING DETAILS

**Reward Model**    We initialize the reward model with the parameters of the SFT model and adopt the pairwise comparison loss (Ouyang et al., 2022) for training. The training is conducted for 1 epoch, with learning rates set to $2e^{-5}$ for the 7B model and $5e^{-6}$ for the 13B model. The weight decay is set to 0. The training size of the reward model is set to 200,000 pairs unless otherwise specified. During inference, the reward model produces scalar outputs to provide the score for the responses.

| Model | Language Model | DCSCORE $\mathcal{F}$ |
|---|---|---|
| Qwen-VL-Chat-7B (Bai et al., 2023) | Qwen-7B | 19.16 |
| mPLUG-Owl2 (Ye et al., 2024) | LLaMA-2-7B | 23.27 |
| LLaVA-1.5-7B (Liu et al., 2024b) | Vicuna-v1.5-7B | 24.50 |
| LLaVA-1.5-13B (Liu et al., 2024b) | Vicuna-v1.5-13B | 25.55 |
| XComposer2.5-7B (Zhang et al., 2024) | InternLM2.5-7B | 29.60 |
| Cambrian-34B (Tong et al., 2024) | Yi-34B | 35.12 |
| LLaVA-1.6-7B (Liu et al., 2024a) | Vicuna-v1.5-7B | 36.21 |
| MiniCPM-Llama3-V-2.5-8B (Yao et al., 2024) | LLaMA-3-8B | 36.36 |
| LLaVA-1.6-13B (Liu et al., 2024a) | Vicuna-v1.5-13B | 37.98 |
| ViLA-40B (Lin et al., 2024) | Yi-34B | 38.02 |
| InternVL-1.5-20B (Chen et al., 2024b) | InternLM2-20B | 39.28 |
| LLaVA-1.6-34B (Liu et al., 2024a) | Yi-34B | 40.46 |
| LLaVA-Onevision-7B (Li et al., 2024a) | Qwen2-7B | 43.49 |
| Gemini-Pro-1.5 (Team et al., 2023) | - | 46.34 |
| InternVL-2-8B (Chen et al., 2024b) | InternLM2.5-7B | 47.39 |
| GPT-4v (OpenAI., 2024b) | - | 48.52 |
| InternVL-2-26B (Chen et al., 2024b) | InternLM2.5-20B | 49.59 |
| GLM-4v-9B (GLM et al., 2024) | GLM-4-9B | 49.85 |
| InternVL-2-40B (Chen et al., 2024b) | Yi-34B | 51.17 |
| Claude-3.5-Sonnet (Anthropic., 2024) | - | 52.37 |
| GPT-4o (OpenAI., 2024a) | - | 53.44 |
| FEEDQUILL-7B | Qwen2-7B | 55.65 |

Table 11: The performance of various VLMs on DECAPBENCH.

**PPO** Our implementation of the PPO algorithm is a variant of (Ouyang et al., 2022). We adopt two reward models: a $c_p$ RM and a $c_r$ RM. The $c_p$ RM is trained with the preference for the proportion of correct units, which measures the precision or hallucination rate of the description of the image. The $c_r$ RM is trained with the preference for the number of primitive information units, which measures the richness of the description of the image. We sum the two RM outputs to a final reward: $r = c_p + \alpha_r c_r$. The hyper-parameter $\alpha_r$ controls the trade-off between accuracy and richness, we set it to 0.5 in our experiments. We set temperature to 1.0 and top-P to 0.7 when sampling trajectories for the diversity of responses. The PPO training data is entirely composed of captioning task data, containing 100k images. Other PPO hyper-parameters are presented in Table 12.

| Hyper-parameter | Default Value |
|---|---|
| Optimizer | AdamW ($\epsilon = 1e - 8$) |
| Learning Rate | 1e-6 (actor), 5e-6 (critic) |
| Scheduler | Linear |
| Batch Size | 256 |
| $\beta$ (KL Penalty Coefficient) | 0.05 |
| $\gamma$ (discount factor) | 1.0 |
| $\lambda$ (TD trade-off factor) | 0.95 |
| Number of Mini-batches | 1 |
| $\epsilon$ (Policy Clipping Coefficient) | 0.2 |
| $\epsilon_v$ (Value Clipping Coefficient) | 0.2 |

Table 12: PPO hyper-parameters

### A.3.2 EVALUATION METRICS AND BENCHMARKS

- MMBench (Liu et al., 2023) introduces a diversity of evaluation questions, and use circular evaluation protocol for multiple choices that leverage GPT to transform free-form answer into the choice.

- MMStar (Chen et al., 2024a) is a vision-critical multi-modal benchmark with 1,500 human-curated challenge samples designed to evaluate 6 core capabilities and 18 detailed axes of VLMs. It is enhanced by strict human review to ensure visual dependency.

- TextVQA (Singh et al., 2019) measures the capability of VLMs for answering question about the text in the natural images.

- VizWiz (Gurari et al., 2018) comes from a natural visual question answering dataset for blinding people.

- ScienceQA (Lu et al., 2022) consists of approximate 21K multi-modal multiple choice questions with a diverse set of science topics and annotations of their answers with corresponding lectures and explanations.

- mmHal-V (Amirloo et al., 2024) is a visual hallucination evaluation benchmarks for VLMs, which consists object attribute, adversarial object, comparison, counting, spatial relation, environment, holistic description, and other types.

- LLaVA-W (Liu et al., 2024b) aims to evaluate the model's capability in visual chatting, which including memes, indoor and outdoor scenes, painting, sketches, etc. Each each image is associated with a highly-detailed and manually-curated description and a proper selection of questions, and utilize GPT to score the model's response.

- WildVision (Lu et al., 2024) simulates the arena and evaluate the model with various real-world questions, while benchmarking human preference.

- CHAIR$_S$ and CHAIR$_I$ (Chan et al., 2023) a widely-recognized tool for evaluating the incidence of object hallucination in image captioning tasks which assess object hallucination at the instance-level and sentence-level respectively.

- MME (Fu et al., 2023) is a comprehensive benchmark for evaluating the capabilities of VLMs in multi-modal tasks. It systematically assesses models across two primary dimensions: perception and cognition, through 14 meticulously designed subtasks that challenge the models' interpretive and analytical skills.

- SeedBench (Li et al., 2024b) consists of 19K multiple choice questions with accurate human annotations, and it spans 12 evaluation dimensions including the comprehension of both the image and video modality.

- MMMU (Yue et al., 2024) includes 11.5K meticulously collected multi-modal questions from college exams, quizzes, and textbooks, covering six core disciplines: Art & Design, Business, Science, Health & Medicine, Humanities & Social Science, and Tech & Engineering.

### A.3.3 PREFERENCE OPTIMIZATION

The following algorithm demonstrates how to leverage PPO (Schulman et al., 2017) to optimize the base model (SFT Model) with reward models trained with preference data $\mathcal{D}$ for $c_p$ and preference data $\mathcal{D}_r$ for $c_r$.

### A.3.4 EVALUATION PROMPT FOR DCSCORE

To measure the quality of the generated captions, we present prompts for decomposition in Table 13, matching in Table 14, and verification in Table 15. We utilize GPT-4o (OpenAI., 2024a) through the whole evaluation process.

### A.3.5 TRAINING PROMPT FOR PPO

We prompt GPT-4o (OpenAI., 2024a) to generate a series of image captioning prompts for PPO training, as listed in Table 16.

---

**Algorithm 1** Preference Optimization with FEEDQUILL

**Input** initial policy model $P_{\theta_{\text{init}}}$; initial value model $V_{\psi_{\text{init}}}$; reward models $R_{\phi_{p/r}}$ trained from $c_p$ or $c_r$; PPO training prompts $\mathcal{D}_t$; PPO hyperparameters $\gamma$, $\lambda$, $\epsilon$, $\beta$.

1: policy model $P_\theta \leftarrow P_{\theta_{\text{init}}}$, value model $V_\psi \leftarrow V_{\psi_{\text{init}}}$
2: **for** step = 1, ..., T **do**
3:     Sample a batch $\mathcal{B}$ from $\mathcal{D}_t$
4:     Sample output sequence $y^n \sim P_\theta(\cdot \mid x^n)$ for each prompt $x^n \in \mathcal{B}$
5:     Compute rewards $\{r_{p_t}^n + r_{r_t}^n\}_{t=1}^{|y^n|}$ from the reward model $R_{\phi_p}$ and $R_{\phi_r}$ for each $y^n$.
6:     Compute advantages $\{A_t\}_{t=1}^{|y^n|}$ and value targets $\{V^{\text{est}}(s_t)\}_{t=1}^{|y^n|}$ for each $y^n$ with $V_\psi$.
7:     **for** PPO iteration = 1, ..., $\mu$ **do**
8:         Update the policy model by maximizing the PPO clipped surrogate objective:

$$\theta \leftarrow \arg\max_\theta \frac{1}{|\mathcal{B}|} \sum_{n=1}^{|\mathcal{B}|} \frac{1}{|y^n|} \sum_{t=1}^{|y^n|} \min\left( \frac{P_\theta(a_t \mid s_t)}{P_{\theta_{\text{old}}}(a_t \mid s_t)} A_t, \text{clip}(v_t, 1-\varepsilon, 1+\varepsilon) A_t \right)$$

9:         Update the value model by minimizing a $L_2$ objective:

$$\psi \leftarrow \arg\min_\psi \frac{1}{|\mathcal{B}|} \sum_{n=1}^{|\mathcal{B}|} \frac{1}{|y^n|} \sum_{t=1}^{|y^n|} \left( V_\psi(s_t) - V^{\text{est}}(s_t) \right)^2$$

10:     **end for**
11: **end for**
**Output** $P_\theta$

---

You are a linguistic expert in extracting primitive information units in the image caption. In specific, "primitive information units" refer to the smallest standalone pieces of information that collectively represent the entire meaning of the sentence without losing any detail, which typically describe various properties of the visual elements in an image. The primitive information unit should be a simple statement. The fact must represent the smallest piece of information that cannot be further broken down without loss of meaning. Abstract concepts or broad interpretations should be reduced to more basic, constituent observations if possible. The primitive information unit should only contain ONE primary element.

When extracting primitive information units from image caption, it is useful to assign unique identifiers to the primary objects or entities being discussed. This will help in maintaining clarity and preventing confusion, especially when there are multiple similar objects or entities. For example, if the caption mentions two cats, you can assign unique identifiers such as "cat$_1$" and "cat$_2$" to distinguish them. Besides, for each attribute, you should also assign the identifier to the object it belongs to. Meanwhile, for spatial relationships, you can assign the identifier to the object that is the subject of the relationship in the primitive information unit.

For each primitive information unit, you should also need to justify whether the primitive information unit directly describe the image or not.

**IMPORTANT**: Please extract ALL of the primitive information units in the image caption. DO NOT omit any information!

The output should be a list of dict [{"fact": [PRIMITIVE INFORMATION UNIT], "identifier": [UNIQUE ID], "relevance": 1/0}, ...] into JSON format. The "identifier" would be optional, if the item in the fact has already been identified with ids. The "relevance" would be 1 if the primitive information unit directly describe the content of the image. Otherwise it would be 0 if the primitive information unit is inference or extension to the description and not directly describe to the content of image.
>>> Caption: {Caption Here}

Table 13: The prompt for decomposing the generated captions into set of primitive information units.

You are now a visual-linguistic expert in matching two set of primitive information units generated from two captions.

You will be received a set of predicted primitive information units across a variety of categories and a set of oracle primitive information units (ground truth). The set of primitive information units is represented as a list of dict [{"fact": [PRIMITIVE INFORMATION UNIT], "identifier": [UNIQUE ID]}, ...] within JSON format. In addition, each primitive information unit in the oracle set would be accompanied with a unique "id" to identify the oracle primitive information unit.

To match primitive information units from a predicted set in terms of the given image with oracle set of primitive information units. Here is the step by step instruction:
1. Preliminary Review: Conduct an initial review of both sets of primitive information units, considering all primitive information units. Understand the details and context presented within each primitive information unit.
2. Inferring Identifier Mappings: Closely examine both sets to deduce potential correlations and mappings based on the content of the primitive information units. Determine if there are any unique identifiers or descriptors that hint at matching entities between the sets. For example, "$cat_0$" in the predicted set's primitive information units may be mapped to "$cat_1$" in the oracle set's primitive information units. Consider the attribute and spatial relation in both sets for possible mapping. Please note that there might be some attribute and spatial errors when mapping the objects. Try find the most similar mapping if exists (not need exact matching). If no oracle primitive information unit matches, simply set matched oracle id to "None".

**IMPORTANT**: Please consider each primitive information unit in the set individually, and MUST NOT omit any primitive information units from the predicted set.

You should only output the matching results which will be formatted as a list of dict as [{"fact": [PRIMITIVE INFORMATION UNIT], "identifier": [UNIQUE ID], "matched_oracle_id": [CORRE-SPONDING ORACLE ID]}, ...] in JSON format. The "identifier" would be optional, if the item in the fact has already been identified with ids as illustrated in the predicted primitive information units. For key named "matched_oracle_id", the value of "matched_oracle_id" should be the corresponding "id" of the primitive information unit in the oracle set. For the primitive information unit in the predicted set which cannot be matched with any oracle primitive information unit, set the value of "matched_oracle_id" to "None".

> > > Set of Primitive information units: {set of units for generated caption}

> > > Oracle Set of Primitive information units: {set of units for human-written caption}

> > > Matching Result:

Table 14: The prompt for verifying the correctness of each primitive information units by utilizing both image and human-written caption.

You are an extraordinary visual-linguistic expert in verifying the correctness of a set of primitive information units given the image and the corresponding reference caption. The set of primitive information units are extracted from a paragraph of machine-generated image caption of that image.

The set of primitive information units is represented as a list of dict ["fact": [PRIMITIVE INFORMATION UNIT], "identifier": [UNIQUE ID], ...] within JSON format. The identifier is unique and to identify the primary objects or entities being discussed. This will help in maintaining clarity and preventing confusion, especially when there are multiple similar objects or entities. For example, if the caption mentions two cats, we would assign unique identifiers such as "cat$_1$" and "cat$_2$" to distinguish them. Besides, for each attribute, it also assigned the identifier to the object it belongs to. Meanwhile, for spatial relationships, it assigned the identifier to the object that is the subject of the relationship in the primitive information unit.

You should first go through all of the primitive information units, and understand the details and context presented within each primitive information unit. Then you need to verify the correctness of each individual primitive information units by asking yourself: Statement: "[PRIMITIVE INFORMATION UNIT]" Does the statement correct according to image or reference caption?

The output for the predicted set of primitive information units should be formatted as a list of dict as ["fact": [PRIMITIVE INFORMATION UNIT], "identifier": [UNIQUE ID], "verification": 1/0, ...] in JSON format, where 1 represents the fact is correct and 0 represents the fact is incorrect. Other keys in the dictionary are the same as the input. The "identifier" would be optional, if the item in the fact has already been identified with ids as illustrated in the input.

> > > Reference Caption: {reference caption}

> > > Primitive Information Units: {primitive information units}

Table 15: The prompt for verifying the correctness of each primitive information units by utilizing both image and human-written caption.

- What do you see happening in this image?
- Can you describe what is happening in this picture?
- What events are taking place in this image?
- What do you observe in this photo?
- Can you explain the scene depicted in this image?
- What is this photo about?
- What is the subject of this picture?
- Can you explain the theme of this image?
- What is the focus of this photo?
- What is the central topic of this picture?
- What is the main idea of this image?
- What is the essence of this photo?
- What is the core subject of this picture?
- What is the primary focus of this image?
- What is the overall theme of this photo?
- What is the main topic depicted in this picture?
- Can you elaborate on the elements of the picture provided?
- Can you give more details about the components of this image?
- What are the various elements in this picture?
- Can you describe the different parts of this photo?
- What are the individual components of this image?
- Can you break down the elements of this picture?
- What are the distinct features of this photo?
- Can you provide more information on the elements in this image?
- What are the specific parts of this picture?
- Can you detail the elements present in this photo?
- are the various aspects of this image?
- Analyze the image in a comprehensive and detailed manner.
- Provide a thorough analysis of this picture.
- Can you give an in-depth examination of this image?
- What is your detailed analysis of this photo?
- Can you break down this image comprehensively?
- What is your extensive analysis of this picture?

Table 16: Part of example prompts for preference optimization.

