# OpenReview forum: "Painting with Words: Elevating Detailed Image Captioning with Benchmark and Alignment Learning"
_ICLR.cc/2025/Conference — ICLR 2025 Poster_

### Official Review · Reviewer_t2Sb · 2024-10-19

**Soundness:** 4
**Presentation:** 4
**Contribution:** 3
**Rating:** 6
**Confidence:** 4

**Summary:**

This work introduces a specialized metric (**DCScore**) and a benchmark (**DeCapBench**) for detailed image description evaluation. The core idea is to break down the reference and generated caption into the "smallest self-sufficient units", and then quantify the precision and recall of information units conveyed by the generated caption. The authors demonstrate that the new metric and benchmark achieve the best consistency with human evaluations. In addition, based on a similar concept, the authors propose a method (**FeedQuill**) for automatically constructing preference data for RLHF. Extensive experiments validate that the collected preference data can train a strong image captioning model.

**Strengths:**

1. This paper introduces a new metric and benchmark for evaluating the quality of detailed image captions. Correlation analysis with human evaluations indicates that these new assessments are effective and superior.
2. This paper presents an efficient method for collecting preference data and demonstrates that such data can be used to build more effective image captioning models.
3. The experiments are comprehensive. The authors’ claims are well-supported by substantial experimental evidence, and they have conducted detailed ablation studies, providing valuable best practices for the community.
4. The paper is well-written with clear figures and tables, effectively conveying information.

**Weaknesses:**

1. Section 3.2 does not introduce the instructions provided to human annotators for scoring image captions. Disclosing the task instruction for annotators is crucial; if the basis for scoring largely aligns with the design of an automated metric, the metric will likely benefit in correlation assessments.
2. DCScore relies on the hyper-detailed human captions in the ImageInWords dataset. However, as "a picture is worth a thousand words," reference descriptions might not fully reflect all the semantics of an image, while the model may describe image details that are correct but not mentioned in the reference caption.
3. The proposed metric conceptually resembles prior works like FaithScore and RLAIF-V (I am delighted to see this discussed in Appendix). The divide-and-conquer approach and evaluation using LLMs is not novel. Collecting preference data for model optimization is also a consensus in the research community. While I see no other obvious flaws, **I am not fully convinced of the overall contribution**. I look forward to being further convinced by the authors and other reviewers.

**Questions:**

1. The main experiments employ LLaVA-Onevision-7B. Why was this setting not maintained consistently in other experiments? For instance, the ablation study “Preference Data for Reward Model” used LLaVA-1.5-7B, and the “Source of Response” experiments used LLaVA-1.5-13B.
2. In Appendix A.1.1, the authors claim that DCScore accounts for non-descriptive elements, unlike other metrics. Could the authors further explain why it is important to consider **non-descriptive** elements in **image captioning** tasks, which aim to generate descriptions for images?

---

> ### Author Response · Authors · 2024-11-22
>
> > **Q1.** Human annotators selection & annotation criteria
>
> **A1.** For the selection of human annotators, we chose individuals with high expertise and language proficiency to ensure objectivity and impartiality. Annotators were familiarized with the caption scoring task but were blinded to the specific study objectives to avoid bias.
>
> Regarding the annotation criteria, detailed guidelines were provided to ensure consistency. Each caption was scored on a scale of 0-4, and the average of three annotators' scores was used as the final score. The scoring criteria are as follows:
>
> - 4: Comprehensive and accurate caption with no key information missed and error-free language.
> - 3: Caption meets main requirements with minor issues like typos or missing background elements.
> - 2: Caption meets main requirements but has significant errors such as hallucinations or positional inaccuracies.
> - 1: Caption has many problems but is slightly better compared to a 0, with errors not exceeding 70%.
> - 0: Caption has severe hallucinations, serious grammatical errors, and confused logic, making it unusable.
>
> We will include the process of human annotator selection and criteria for annotating the caption in the revised manuscript.
>
> > **Q2.** Omission captioning details in ground-truth annotation
>
> **A2.** As we explained to Reviewer DgKk, we compensate for the missing fine-grained details in the ground-truth captions that are present in the model's predictions when computing recall score $s_r$ for DCScore. In concrete, in step 3, we would use GPT-4o to identify the units that are correct but not presented in reference captions. By incorporating the correct additional information from the model's captions into the ground-truth caption, we make the evaluation metric more robust to the omission details in ground-truth captions.
>
> | Considering Omission in Ground-truth  | PCC ($\rho$) | $1-R^2$ | KD $\tau$ |  Sp $\tau$ |
> |---|---|---|---|---|
> | No | 0.6151 | **0.72** | 0.5111 | 0.5916 |
> | Yes (DCScore) | **0.6605** | 1.54 | **0.5328** | **0.6166** |
>
>
>
> > **Q3.** Restatement of the contribution
>
> **A3.** Our main contributions are summarized as follows:
> - **Novel Image Captioning Metric and Benchmark**: We present DeCapBench alongside the DCScore metric designed specifically to evaluate detailed image captioning tasks. Unlike current mainstream captioning evaluation datasets that tend to focus on short-caption evaluations, DeCapBench is developed to handle the complexity and richness of detailed image captions. Current evaluation methods often overlook the detailed context and potential hallucinations present in longer, more descriptive captions. By providing a metric that assesses both the quality and hallucination degree of these detailed captions, we fill a significant gap identified by Reviewer Mger.
> - **New Fine-grained Feedback Collection Method**: While generating preference data is a consensus in the research community, the methodology to collect such data significantly influences the accuracy of preference pairs. Therefore, we introduce FeedQuill to automatically collect high-quality preference data by considering both hallucination and richness in the generated captions. This method is scalable, as highlighted by Reviewer A3YP. Additionally, the preference data collected by FeedQuill generalizes much better than other methods, as supported by Reviewer DgKk.
> - **Comprehensive Experiments**: To demonstrate the effectiveness and generalization ability of the proposed FeedQuill, we provide extensive experiments on a diverse range of downstream tasks and across different types of VLMs. The results show reduced hallucinations, superior performance in visual chat compared to GPT-4v, and better detailed image captioning capabilities than GPT-4o.

---

> ### Author Response · Authors · 2024-11-22
>
> > **Q4.** Experimental settings
>
> **A4.** For the main experiments, we chose LLaVA-Onevision-7B because it is the most capable model recently available, allowing us to showcase the maximum potential of our method. To demonstrate the generalization capability of our proposed method, we also included a variety of VLMs in different experiments, as shown in Table 6. In the ablation study "Preference Data for Reward Model," we utilized LLaVA-1.5-7B to ensure a fair comparison with other models and datasets, maintaining consistency with common baselines. For the "Source of Response" experiments, we employed LLaVA-1.5-13B to ensure diversity and robustness by using responses from both weaker and stronger VLMs. This varied selection helps us better understand how our method performs across different contexts and strengths, providing a comprehensive evaluation of its robustness and generalizability.
>
> > **Q5.** Motivation of including non-descriptive captions
>
> **A5.** We investigated the influence of non-descriptive elements in DCScore on its alignment with human judgment, as follows:
>
> | Including Non-Descriptive Elements | PCC ($\rho$) | $1-R^2$ | KD $\tau$ |  Sp $\tau$ |
> |---|---|---|---|---|
> | No | 0.6213 | 2.77 | 0.5048 | 0.5985 |
> | Yes (DCScore) | **0.6605** | **1.54** | **0.5328** | **0.6166** |
>
> The results show that including non-descriptive elements during detailed image caption evaluation achieves a higher correlation with human judgment. This improvement occurs because non-descriptive elements, such as background details and inferred information, provide additional context that leads to a more comprehensive understanding of the image content. Consequently, by including these elements, DCScore captures subtle nuances and implicit information critical for fully understanding the image, thus more closely aligning with human judgment.
>
> Therefore, incorporating non-descriptive elements into DCScore provides a more accurate and reliable evaluation of generated captions, addressing a gap in previous metrics that have overlooked these contextual details.

---

> ### Comment · Reviewer_t2Sb · 2024-11-22
> **Response to Authors**
>
> Thank you for your detailed response.
>
> For Q1 and Q2, my concerns have been addressed. Regarding Q3, I am now convinced by the overall contribution. Q4 and Q5 were due to my misunderstandings, and I appreciate your clarifications.
>
> This paper provides a thorough exploration of image captioning. It introduces an evaluation framework that effectively bridges the gap between existing metrics and the current status of vision-language research. I am also impressed by the performance of the simple yet effective preference optimization method. I believe this work makes a solid contribution to the field and will be beneficial for the ongoing development of vision-language models. I will increase my rating to a 6.

---

> ### Comment · Area_Chair_FdcN · 2024-11-27
>
> Dear reviewer,
>
> Today is the last day for reviewers to ask questions to authors. Did the authors' rebuttal address your concern? Do you have any additional questions?

---

> > ### Comment · Reviewer_t2Sb · 2024-11-27
> > **Request for Confirmation of Reviewer Feedback Visibility**
> >
> > Dear Authors and Area Chair,
> >
> > I am writing to seek clarification regarding the visibility of my feedback provided on 22 November. Below is a summary of the responses I provided:
> >
> > ```
> > Thank you for your detailed response.
> >
> > For Q1 and Q2, my concerns have been addressed. Regarding Q3, I am now convinced by the overall contribution. Q4 and Q5 were due to my misunderstandings, and I appreciate your clarifications.
> >
> > This paper provides a thorough exploration of image captioning. It introduces an evaluation framework that effectively bridges the gap between existing metrics and the current status of vision-language research. I am also impressed by the performance of the simple yet effective preference optimization method. I believe this work makes a solid contribution to the field and will be beneficial for the ongoing development of vision-language models. I will increase my rating to a 6.
> > ```
> >
> > However, I have noticed that the authors have not responded to them. Additionally, the Area Chair has raised inquiries regarding my feedback, which leads me to question whether my responses are visible to all relevant parties. According to my interface on OpenReview, my comment appears publicly visible to `Everyone`. Could you please confirm if you have received my feedback?
> >
> > Reviewer t2Sb

---

> > > ### Author Response · Authors · 2024-11-27
> > >
> > > Thank you for your detailed feedback and for addressing each of your concerns. We are pleased that our responses effectively resolved your concerns. And we are grateful for your recognition of our work.
> > >
> > > Best regards,
> > >
> > > Authors of 2960

---

### Official Review · Reviewer_swUh · 2024-10-25

**Soundness:** 3
**Presentation:** 3
**Contribution:** 3
**Rating:** 6
**Confidence:** 3

**Summary:**

This paper propose:
1. DCScore: a metric to evaluate both hallucination and comprehensiveness


2. DeCapBench: an image captioning benchmark (contains only testing dataset) for hallucination evaluation


3. FeedQuill: a method to mitigate hallucination in vision-language models (VLMs), which consists of the following steps:
  - (1) Collect the responses from VLMs.
  - (2) Employ LLM to decompose the responses into primitive information units.
  - (3) Use an off-the-shelf VLMs to verify the correctness of each information units.
  - (4) Label data as positive or negative based on the verification scores.
  - (5) Train a reward model with preference dataset constructed in (4).
  - (6) Fine-tune the target VLM to generate less hallucinated and more enriched captions through PPO.

Finally, several vl benchmarks are achieved as SOTA performance.

**Strengths:**

- Mitigate the hallucination issues in VLM is crucial, especially in the detailed image captioning task.
- The proposed metric DCScore sounds reasonable, is provided with a comprehensive comparison with previous metrics (e.g., Faithscore and RLAIF-V), and is demonstrated high consistency with human evaluation.
- The conducted experiments and related ablation studies are extensive.

**Weaknesses:**

[Metric - DCScore]
1. Why are non-descriptive captions included as a part of this metric?
2. This metric appears to rely on paid API (i.e., GPT-4o) for its evaluation process. It would be advantageous if the metric could also be adapted to work with open-source VLMs as alternatives to GPT-4o.

[Benchmark - DeCapBench]
1. The testing dataset in DeCapBench consists of only 400 samples. How does this compare to other visual data hallucination quality sets, such as HallusionBench [1]

[Method - FeedQuill]
1. In table3, there is a lacked experiment to compare the FeedQuill with the simplest cross-entropy loss (i.e., image caption loss) using the same PPO-finetuned dataset. A comparison of FeedQuill with cross-entropy loss on hallucination-measured datasets, such as mmHal-V and DeCapBench, would be valuable.
2. In addition to MSCOCO, OpenImages, and ShareGPT4V, what other datasets are included in fine-tuning the VLM with PPO?
3. Is the preference score $c_r$ a scalar value? If so, why is it necessary to train an additional reward model $R_{\phi_r}$ to generate the $r_{r_t}$ in Algo 1? Could the $c_r$ be directly used as a part of reward?


[1] HallusionBench: An Advanced Diagnostic Suite for Entangled Language Hallucination and Visual Illusion in Large Vision-Language Models (CVPR'24)

**Questions:**

Please answer my questions in “weaknesses” section.

---

> ### Author Response · Authors · 2024-11-22
>
> > **Q1.** Motivation of including non-descriptive captions
>
> **A1.** We investigated the influence of non-descriptive elements in DCScore on its alignment with human judgment, as follows:
>
> | Including Non-Descriptive Elements | PCC ($\rho$) | $1-R^2$ | KD $\tau$ |  Sp $\tau$ |
> |---|---|---|---|---|
> | No | 0.6213 | 2.77 | 0.5048 | 0.5985 |
> | Yes (DCScore) | **0.6605** | **1.54** | **0.5328** | **0.6166** |
>
> The results show that including non-descriptive elements during detailed image caption evaluation achieves a higher correlation with human judgment. This improvement occurs because non-descriptive elements, such as background details and inferred information, provide additional context that leads to a more comprehensive understanding of the image content. Consequently, by including these elements, DCScore captures subtle nuances and implicit information critical for fully understanding the image, thus more closely aligning with human judgment.
>
> Therefore, incorporating non-descriptive elements into DCScore provides a more accurate and reliable evaluation of generated captions, addressing a gap in previous metrics that have overlooked these contextual details.
>
>
> > **Q2.** - Open-sourced alternatives to GPT-4o in evaluation
>
> **A2.** Our evaluation metric, DCScore, can indeed be adapted to open-source VLMs. To demonstrate this, we conducted an experiment using Qwen2-VL, a recently released VLM in the open-source community. We adopted the same evaluation prompts for DCScore without any special tuning and compared the results with those obtained using GPT-4o. The results of using Qwen2-VL in terms of human consistency are as follows:
>
> | Evaluation VLM | PCC ($\rho$) | $1-R^2$ | KD $\tau$ |  Sp $\tau$ |
> |---|---|---|---|---|
> | GPT-4o | 0.6605    | 1.54 | 0.5328 | 0.6166 |
> | Qwen2-VL | 0.5792 | 0.90 | 0.4669 | 0.5340 |
>
> The comparison indicates that while GPT-4o achieves a higher degree of human consistency, Qwen2-VL also performs robustly, demonstrating the flexibility and adaptability of DCScore to different VLMs. Furthermore, Qwen2-VL's performance with DCScore remains superior compared to other traditional metrics, showcasing the metric's robustness and adaptability. Notably, adapting DCScore to open-source models like Qwen2-VL allows for broader accessibility and cost-efficiency, without significantly compromising the reliability of the evaluation process.

---

> ### Author Response · Authors · 2024-11-22
>
> > **Q3.** Sample selection of DeCapBench
>
> **A3.** DeCapBench's images and captions are sourced from ImageInWords [1], which provides only 400 images with high-quality human-written captions. While 400 images may seem like a small subset, these are the only ones released and are highly valuable due to their exceptional quality and the costly, time-intensive process of creating them. Each caption is hyper-detailed and meticulously crafted by well-educated human annotators, taking approximately 1800 seconds to produce [1]. This high quality is crucial for evaluating detailed image descriptions. Our findings in Table 2 show that higher-quality ground-truth captions lead to better alignment with human judgment, underlining the importance of using high-quality captions.
>
> On the other hand, we tested the performance variance when varying the size of DeCapBench. Specifically, we ran the evaluation multiple times and computed the standard deviation of the performance. The results demonstrate that increasing the sample size from 100 to 400 significantly reduces the standard deviation from **1.27 to 0.07**, leading to more stable evaluations. Additionally, we observed that other VLM evaluation benchmarks such as MIABench [2] and HallusionBench [3] employ similar sample sizes, suggesting that a sample size of 400 is a reasonable choice for achieving reliable evaluation outcomes.
>
> [1] Garg, Roopal, et al. "ImageInWords: Unlocking Hyper-Detailed Image Descriptions." arXiv preprint arXiv:2405.02793 (2024).
>
> [2] Qian, Yusu, et al. "Mia-bench: Towards better instruction following evaluation of multimodal llms." arXiv preprint arXiv:2407.01509 (2024).
>
> [3] Guan, Tianrui, et al. "HallusionBench: an advanced diagnostic suite for entangled language hallucination and visual illusion in large vision-language models." Proceedings of the IEEE/CVF Conference on Computer Vision and Pattern Recognition. 2024.
>
>
> > **Q4.** Comparison to HallusionBench
>
> **A4.** HallusionBench is designed for evaluating image-context reasoning. It differs from our DeCapBench in the following aspects:
> 1. **Types of Hallucinations Evaluated**: DeCapBench evaluates visual hallucinations in image captions, focusing on aspects such as object existence, correctness of attributes, and relationships. Visual hallucinations refer to the inclusion of objects, attributes, or relationships that do not exist in the image. In contrast, HallusionBench primarily addresses language hallucinations, which stem from overreliance on language priors rather than visual context. Additionally, HallusionBench measures visual illusions, which denote the misinterpretation of accurate visual information—where an existing object or relationship is incorrectly perceived.
> 2. **Types of Evaluation Tasks**: DeCapBench evaluates the quality of detailed image captions across various aspects, not limited to hallucinations. It assesses the accuracy and richness of the captions. On the other hand, HallusionBench is designed to evaluate the image-context reasoning capability within a QA format, focusing on how well the reasoning aligns with image context rather than the captioning quality.
> 3. **Coverage of Evaluation**: DeCapBench not only evaluates hallucinations in the generated image captions but also assesses comprehensiveness using hyper-detailed human-curated captions. This means it looks at how well the captions cover all relevant details in the image. In contrast, HallusionBench focuses solely on evaluating "language hallucinations" and "visual illusions," specializing in hallucination evaluation without assessing the overall comprehensiveness of responses.
>
> > **Q5.** Fine-tuning with PPO train datasets for Table 3
>
> **A5.** Instead of instruction fine-tuning, our method employs Proximal Policy Optimization (PPO), a reinforcement learning algorithm. During the PPO training process, the model generates responses on-the-fly based on input prompts rather than relying on pre-existing ground-truth annotations. As a result, it is not feasible to directly fine-tune on PPO training data using cross-entropy loss due to the absence of ground-truth annotations.
> For reference, we also present the results of rejection sampling, where the response annotations are generated by selecting the best-of-N responses from the base model. These results are included below:
>
> | Method | MMBench | MMStar | WildVision | mmHal-V | DeCapBench |
> |---|---|---|---|---|---|
> | Base Model | 64.8 | 33.1 | 14.48 | 1.85 | 24.50 |
> | Rejection Sampling (Cross-entropy) | 64.2 | 34.0 | 16.21 | 2.22 | 26.25 |
> | FeedQuill | **66.3** | **35.8** | **19.68** | **2.60** | **34.52** |

---

> ### Author Response · Authors · 2024-11-22
>
> > **Q6.** Data source for PPO training
>
> **A6.** We randomly sample images and caption prompts from the datasets listed in Table 14 of the Appendix. The image sources include MSCOCO, OpenImages, and ShareGPT4V; no annotated responses are used.
>
> > **Q7.** Necessity of reward model $R_{\phi_r}$
>
> **A7.** The preference score $c_r$ is indeed a scalar, produced by counting primitive information units decomposed by the model's responses using an LLM. Directly integrating response decomposition for primitive information units with LLM generation into PPO training, although potentially more accurate, is time-consuming. The time complexity for LLM generation is O(N), while that for the reward model is O(1). Hence, we train the reward model $R_{\phi_r}$ to substitute the direct generation of $c_r$ for each response, improving efficiency in the training process.
> Furthermore, we have evaluated the accuracy of the reward model $R_{\phi_r}$ in predicting the comparative relationship of $c_r$. It achieves an accuracy of 96.3% on pairwise comparisons, demonstrating its reliability in the training process. This high accuracy indicates that $R_{\phi_r}$ can effectively approximate the preference scores $c_r$, ensuring that the training process remains both efficient and accurate.

---

### Official Review · Reviewer_DgKk · 2024-10-26

**Soundness:** 3
**Presentation:** 3
**Contribution:** 3
**Rating:** 6
**Confidence:** 4

**Summary:**

This paper introduces a new metric DCScore and a new benchmark DECAPBENCH to evaluate the detailed image captioning capabilities of  VLMs. DCScore is designed to measure both the hallucination and comprehensiveness of generated captions. To calculate DCScore, ground-truth and generated captions are first broken down into primitive information units. Then, the primitive information units from the generated captions are compared with those from the ground-truth captions. In addition, GPT-4o is utilized to judge whether each primitive information unit from the generated captions corresponds to the image. Based on these results, a precision score and a recall score are derived, representing non-hallucination and comprehensiveness, respectively. Empirical study shows that DCScore is more aligned with human judgments than previous image captioning evaluation metrics. By combining the proposed DScore and 400 high-quality and detailed image-caption pairs, the benchmark DECAPBENCH is established.
The paper also proposes FEEDQUILL, an automatic fine-grained feedback collection method to collect preference data for model training. This method breaks down each response into primitive information units, ensembles multiple VLMs to score these units, and constructs preference data from these scores. Experiments demonstrate that models trained using FEEDQUILL outperform those trained with other preference data.

**Strengths:**

1. A novel metric for evaluating detailed image captions and an automatic feedback collection method are proposed. The proposed metric aligns more closely with human judgments and can measure hallucination and comprehensiveness. The proposed feedback collection method is able to construct better preference data without human annotators than previous preference data collection methods.
2. The experiments show that the proposed FEEDQUILL generalizes much better than other preference data when LLaVA models are used. In addition, the model trained with FEEDQUILL has better performance on various downstream benchmarks, demonstrating its effectiveness in enhancing models' image captioning capabilities.
3. The paper is well-written and organized.

**Weaknesses:**

1. The explanation of the DCScore evaluation process is not entirely clear. Please see the questions below.
2. When evaluating the effectiveness of FEEDQUILL with various VLMs, only LLaVA-family models are utilized. Why aren't any non-LLaVA-family models (e.g. InternVL-2-8B) used in Table 6?

**Questions:**

About DCScore
Step 1 of the evaluation process is unclear to me.
1. Who are the “human experts”? What’s the definition of “experts” in this paper?
2. Why are the decomposers for generated captions and ground-truth captions different (LLM vs. human experts)? Can LLM be used for both?

Step 3
3. Is the goal of this step to compensate for the missing details in the ground-truth captions?
What’s the difference between $P_{true}$ and $Q$?

About DECAPBENCH
4. How are the 400 high-quality, human-curated public detailed captions chosen? Is there any criterion for this selection?

About FEEDQUILL
5. In the related work section, the paper mentioned that using GPT-4v to collect preference data could pose risks of bias and unreliability as the preference judgment of GPT-4v is not manually verified. As FEEDQUILL also leverages multiple VLMs to collect preference pairs, aren't the collected data also likely to be influenced by these models' bias and unreliability?

About experiments
6. Do other non-LLaVA VLMs, e.g. InternVL-2-8B, trained with the FEEDQUILL-collected preference data also show superior results on downstream tasks?
7. How many FEEDQUILL preference data are used for training in the last row of Table 2?

Minor comments
8. Line 319: "..., responses with fewer hallucinations are often inherently less helpful." Is this sentence correct?
9. Typo in line 334: "In To fully exploit the characteristics ..."

---

> ### Author Response · Authors · 2024-11-22
>
> > **Q1.** Human annotators selection & annotation criteria
>
> **A1.** For the selection of human annotators, we chose individuals with high expertise and language proficiency to ensure objectivity and impartiality. Annotators were familiarized with the caption scoring task but were blinded to the specific study objectives to avoid bias.
>
> Regarding the annotation criteria, detailed guidelines were provided to ensure consistency. Each caption was scored on a scale of 0-4, and the average of three annotators' scores was used as the final score. The scoring criteria are as follows:
>
> - 4: Comprehensive and accurate caption with no key information missed and error-free language.
> - 3: Caption meets main requirements with minor issues like typos or missing background elements.
> - 2: Caption meets main requirements but has significant errors such as hallucinations or positional inaccuracies.
> - 1: Caption has many problems but is slightly better compared to a 0, with errors not exceeding 70%.
> - 0: Caption has severe hallucinations, serious grammatical errors, and confused logic, making it unusable.
>
> We will include the process of human annotator selection and criteria for annotating the caption in the revised manuscript.
>
> > **Q2.** Decomposer for generated captions & ground-truth captions
>
> **A2.** It is feasible to use the same decomposers for both generated and ground-truth captions. However, we opted to utilize human experts for decomposing ground-truth captions for two main reasons:
> 1. **Accuracy**: Using an LLM to decompose ground-truth captions might introduce minor errors.
> 2. **Efficiency**: Ground-truth captions are constant, allowing human experts to decompose them only once, which is less costly.
>
> Additionally, we have tested using an LLM to decompose ground-truth captions and observed that it achieved a Pearson correlation of **0.9279** with decompositions performed by human experts in terms of DCScore. Meanwhile, using both LLM decomposed in model and ground-truth captions, the correlation is demonstrated as follows, which shows the superiority of using human experts for decomposing ground-truth captions.
>
> | Decomposer for Ground-truth | PCC ($\rho$) | $1-R^2$ | KD $\tau$ |  Sp $\tau$ |
> |---|---|---|---|---|
> | LLM | 0.6338 | **0.91** | 0.5179 | 0.6014 |
> | Human Experts | **0.6605** | 1.54 | **0.5328** | **0.6166** |
>
> > **Q3.** Functionality of step 3 in DCScore
>
> **A3.** To clarify, $\mathcal{P}_{true}$ represents the set of all correct primitive information units in the predicted caption, while $\mathcal{Q}$ denotes the common set of correct primitive information units found in both the predicted caption and the ground-truth caption. Therefore, the goal of this step is to compensate for the missing fine-grained details in the ground-truth captions that are present in the model's predictions. Human annotations often omit some of these details, and by incorporating the correct additional information from the model's captions into the ground-truth caption, we make the evaluation metric more robust to these omissions.
>
> | Considering Omission in Ground-truth  | PCC ($\rho$) | $1-R^2$ | KD $\tau$ |  Sp $\tau$ |
> |---|---|---|---|---|
> | No | 0.6151 | **0.72** | 0.5111 | 0.5916 |
> | Yes (DCScore) | **0.6605** | 1.54 | **0.5328** | **0.6166** |

---

> ### Author Response · Authors · 2024-11-22
>
> > **Q4.** Sample selection of DeCapBench
>
> **A4.** DeCapBench's images and captions are sourced from ImageInWords [1], which provides only 400 images with high-quality human-written captions. While 400 images may seem like a small subset, these are the only ones released and are highly valuable due to their exceptional quality and the costly, time-intensive process of creating them. Each caption is hyper-detailed and meticulously crafted by well-educated human annotators, taking approximately 1800 seconds to produce [1]. This high quality is crucial for evaluating detailed image descriptions. Our findings in Table 2 show that higher-quality ground-truth captions lead to better alignment with human judgment, underlining the importance of using high-quality captions.
>
> On the other hand, we tested the performance variance when varying the size of DeCapBench. Specifically, we ran the evaluation multiple times and computed the standard deviation of the performance. The results demonstrate that increasing the sample size from 100 to 400 significantly reduces the standard deviation from **1.27 to 0.07**, leading to more stable evaluations. Additionally, we observed that other VLM evaluation benchmarks such as MIABench [2] and HallusionBench [3] employ similar sample sizes, suggesting that a sample size of 400 is a reasonable choice for achieving reliable evaluation outcomes.
>
> [1] Garg, Roopal, et al. "ImageInWords: Unlocking Hyper-Detailed Image Descriptions." arXiv preprint arXiv:2405.02793 (2024).
>
> [2] Qian, Yusu, et al. "Mia-bench: Towards better instruction following evaluation of multimodal llms." arXiv preprint arXiv:2407.01509 (2024).
>
> [3] Guan, Tianrui, et al. "HallusionBench: an advanced diagnostic suite for entangled language hallucination and visual illusion in large vision-language models." Proceedings of the IEEE/CVF Conference on Computer Vision and Pattern Recognition. 2024.
>
> > **Q5.** Difference of preference data collection method and reliance
>
> **A5.** To address the issues of bias and unreliability in existing preference data collection methods like VLFeedback [4] and LLaVA-Hound [5], our approach differs in the following aspects:
> 1. **Approach of Generating Preference**: VLFeedback and LLaVA-Hound generate preference signals by presenting two candidate responses simultaneously and directly asking the VLM to choose the preferred response. This approach relies on the VLM's holistic judgment, which can introduce biases. In contrast, our method decomposes each response into several primitive information units and uses VLMs to verify the correctness of each unit separately. We then aggregate the fraction of correct units as a score to form the preference pairs. This decomposition and verification mechanism reduces the risk of bias and unreliability by focusing on smaller, verifiable units of information rather than a single, holistic judgment.
> 2. **Reliability of Generated Preference**: Directly prompting VLM for holistic preference judgment by giving two responses simultaneously is susceptible to language preference or positional biases, as demonstrated in [6]. In contrast, by breaking down responses into smaller units and verifying each unit separately, our approach adds an extra layer of granularity and rigor to the verification process. This ensures that the preference pairs formed are based on more precise and reliable judgments, thus mitigating the risk of bias and unreliability that might stem from using a single model's overall preference judgment. To further demonstrate this, we manually annotated 156 preference pairs for validation. The results demonstrate that our verification-based method achieves **88.5%** accuracy while the VLM holistic preference-based method only achieves 58.97%, highlighting the reliability of our preference collection approach.
>
> [4] Li, Lei, et al. "VLFeedback: A Large-Scale AI Feedback Dataset for Large Vision-Language Models Alignment." arXiv preprint arXiv:2410.09421 (2024).
>
> [5] Zhang, Ruohong, et al. "Direct Preference Optimization of Video Large Multimodal Models from Language Model Reward." arXiv preprint arXiv:2404.01258 (2024).
>
> [6] Shi, Lin, et al. "Judging the judges: A systematic investigation of position bias in pairwise comparative assessments by llms." arXiv preprint arXiv:2406.07791 (2024).

---

> ### Author Response · Authors · 2024-11-22
>
> > **Q6.**. Motivation of LLaVA-series models for experiments and performance on other VLMs
>
> **A6.** To demonstrate the effectiveness of our proposed method, we have applied it to InternVL2-8B model with preference optimization. The results are presented as follows:
> | Model | AI2D | MMStar | LLaVA-W | MMVet | DeCapBench |
> |---|---|---|---|---|---|
> | InternVL2-8B | 83.8 | 59.4 | 84.5 | 60.0 | 45.55 |
> | FeedQuill (InternVL2-8B) | **83.9** | **59.4** | **90.5** | **62.5** | **51.57** |
>
> As we can observe in the table, our method still achieves performance gain on several benchmarks, showing the generalization of our method.
> We want to further emphasize that we chose the LLaVA series (1.5/1.6/OneVision) as the base models because they encompass a divserse range of VLMs, differing in terms of **(1)** training data composition, **(2)** image processing strategy (Pad / AnyRes), **(3)** vision backbones (CLIP / SigLIP), and **(4)** LLM backbones (Vicuna / Qwen2). This diversity allows us to comprehensively evaluate the effectiveness and wide applicability of our proposed methodology. Moreover, these different LLaVA models are trained under varying settings, resulting in significantly different capabilities: LLaVA-OneVision is the latest and most capable model in the LLaVA series, comparable to InternVL2-8B [7], while LLaVA-1.6 is comparable to VILA-7B [8].
>
> [7] Chen, Zhe, et al. "Internvl: Scaling up vision foundation models and aligning for generic visual-linguistic tasks." Proceedings of the IEEE/CVF Conference on Computer Vision and Pattern Recognition. 2024.
>
> [8] Lin, Ji, et al. "Vila: On pre-training for visual language models." Proceedings of the IEEE/CVF Conference on Computer Vision and Pattern Recognition. 2024.
>
>
> > **Q7.** Minor Questions
>
> **A7.** We address these minor questions in the following and will present them more clear in the revised manuscript.
> - Preference Data Size: We use 200k preference pairs for the last row of Table 2 as stated in the Appendix A.2.1.
> - Statement in Line 319: Yes, the short responses are less-informative, but tend to have fewer hallucinations. Lengthy responses have a higher probability of involving hallucination.
> - Typos: Thanks for pointing out. We will correct the typos (e.g. Line 334) in the revised manuscript.

---

### Official Review · Reviewer_Mger · 2024-10-26

**Soundness:** 3
**Presentation:** 3
**Contribution:** 3
**Rating:** 6
**Confidence:** 4

**Summary:**

This paper proposes a DeCapBench together with a DCScore and FeedQuill preference optimization method to evaluate and improve the ability of detailed image captioning. More specifically, DCScore is implemented in an F1 score style to asses the hallucination and comprehensiveness of the output captions. The introduced detailed captioning benchmark DeCapBench is further conducted to evaluate the captioning capability of VLMs. In addition, this paper also proposes a fine-grained feedback collection method to formulate the reward function for model alignment. The experimental results demonstrate extensive comparisons and results against multiple closed- and open-sourced approaches.

**Strengths:**

1. Evaluating detailed image captioning remains a challenging task since current mainstream captioning datasets (e.g., COCO) only contain relatively coarse-grained and short captions. Since most of the metrics for image captioning rely on ground-truth captions, designing a metric to properly assess the quality and hallucination degree of the output captions is crucial, especially in the multimodal LLM era.
2. The proposed fine-grained feedback as the reward function and the PPO-based alignment framework is reasonable and technically sound.
3. The quantitative comparisons and experimental analysis are comprehensive, and the performance of the proposed method is promising.
4. The overall paper is well-structured and easy to follow.

**Weaknesses:**

1. As we know, the multimodal LLMs themself has inevitable hallucination issues. Does integrating VLMs into the verification step guarantee that the evaluation results are trustworthy? Or the evaluation results may still be affected by potential hallucinations or input prompts.
2. While this paper provides results like Table 1 to show the proposed DCScore better aligns with human judgments, it remains unclear whether this behavior can be generalized to other datasets or tasks.

**Questions:**

Please refer to the Weaknesses. The following is a minor question.

1. This paper mainly considers LLaVA to be the VLM. Are other commonly used multimodal LLMs, such as VILA or InternVL-2, also applicable to this proposed RL alignment framework?

---

### Official Review · Reviewer_A3YP · 2024-11-03

**Soundness:** 3
**Presentation:** 2
**Contribution:** 3
**Rating:** 6
**Confidence:** 4

**Summary:**

Image captioning is an important task that has recently been explored using VLMs to generate detailed captions. Traditional metrics or coarse annotations may not be ideal to evaluate the performance of detailed image captioning.

This paper proposes evaluation metric for detailed captioning tasks, considering both hallucination and comprehensiveness. A public benchmarks has been proposed. The paper also introduces FEEDQUILL, a scalable method for fine-grained feedback collection by decomposing and verifying responses. Experimental results seem reasonable.

**Strengths:**

To address the problem of evaluating the performance of detailed image captioning, this paper proposes a new evaluation metric to take both hallucination and comprehensiveness into consideration. It also constructed an evaluation benchmark using the proposed evaluation metric to the ImageInWords images and their corresponding hyper-detailed image captions. The experimental results seem reasonable.

**Weaknesses:**

To generate benchmark for detailed image captioning, 400 images from ImageInWords dataset are used to generate benchmark with the proposed evaluation metric. Only 400 images seems a very small subset. The uniqueness of the proposed benchmarks needs to be further clarified.

The performance of the proposed method does not always achieve the best results. More explanations and justifications are expected.

The organisation of the paper can be further improved. It would be good to have a self-contained version rather than leave some important content in appendix.

Some notations are not clearly defined. For example, in 4.1, the definition of the fraction of correct units seems not easy to be understand.

**Questions:**

To generate benchmark, why only 400 images are selected? How these images are selected? 400 images seems very small subset.

Why LLaVA models are used as base model? It will be good if more popularly used models can be investigated to demonstrate the effectiveness of the proposed fine-grained feedback collection.

The decomposition in section 4.1 seems similar as that in 3.1.

---

> ### Comment · Area_Chair_FdcN · 2024-11-27
>
> Dear reviewer,
>
> Today is the last day for reviewers to ask questions to authors. Did the authors' rebuttal address your concern? Do you have any additional questions?

---

### Meta-Review · Area_Chair_FdcN · 2024-12-23

**Metareview:**

This paper was reviewed by 5 experts in the field. The authors' rebuttal resolved most of the concerns, and reviewers unanimously agreed to accept the paper.

The AC agrees with the reviewers' assessments and does not find strong reasons to overturn the reviewers' consensus. The decision is to recommend the paper for acceptance. The reviewers did raise some valuable suggestions in the discussion that should be incorporated in the final camera-ready version of the paper. The authors are encouraged to make the necessary changes to the best of their ability.

**Additional Comments On Reviewer Discussion:**

Reviewer A3YP didn't participate in the discussion despite multiple reminders. The authors' rebuttal successfully addressed most of the concerns from the reviewers. After rebuttal, reviewer Mger and swUh kept their ratings of 6; reviewer DgKk and t2Sb increased their ratings to 6.

---

### Decision · Program_Chairs · 2025-01-22

Accept (Poster)